# The generation and use of recombinant extracellular vesicles as biological reference material

Edward Geeurickx [1,2], Joeri Tulkens[1,2], Bert Dhondt [1,2,3], Jan Van Deun [1,2], Lien Lippens[1,2,4], Glenn Vergauwen[1,2,5], Elisa Heyrman[1], Delphine De Sutter[6,7], Kris Gevaert[2,6,7], Francis Impens [2,6,7,8], Ilkka Miinalainen[9], Pieter-Jan Van Bockstal [10], Thomas De Beer[10], Marca H.M. Wauben[11], Esther N.M. Nolte-'t-Hoen[11], Katarzyna Bloch[12], Johannes V. Swinnen [12], Edwin van der Pol[13,14,15], Rienk Nieuwland[13,15], Geert Braems[5], Nico Callewaert[2,6,7], Pieter Mestdagh[2,6,16], Jo Vandesompele [2,6,16], Hannelore Denys[2,4], Sven Eyckerman[2,6,7], Olivier De Wever[1,2] & An Hendrix [1,2]

Recent years have seen an increase of extracellular vesicle (EV) research geared towards biological understanding, diagnostics and therapy. However, EV data interpretation remains challenging owing to complexity of biofluids and technical variation introduced during sample preparation and analysis. To understand and mitigate these limitations, we generated trackable recombinant EV (rEV) as a biological reference material. Employing complementary characterization methods, we demonstrate that rEV are stable and bear physical and biochemical traits characteristic of sample EV. Furthermore, rEV can be quantified using fluorescence-, RNA- and protein-based technologies available in routine laboratories. Spiking rEV in biofluids allows recovery efficiencies of commonly implemented EV separation methods to be identified, intra-method and inter-user variability induced by sample handling to be defined, and to normalize and improve sensitivity of EV enumerations. We anticipate that rEV will aid EV-based sample preparation and analysis, data normalization, method development and instrument calibration in various research and biomedical applications.

[1] Laboratory of Experimental Cancer Research, Department of Human Structure and Repair, Ghent University, Ghent 9000, Belgium. [2] Cancer Research Institute Ghent, Ghent 9000, Belgium. [3] Department of Urology, Ghent University Hospital, Ghent 9000, Belgium. [4] Department of Medical Oncology, Ghent University Hospital, Ghent 9000, Belgium. [5] Department of Gynaecology, Ghent University Hospital, Ghent 9000, Belgium. [6] Department of Biomolecular Medicine, Ghent University, Ghent 9000, Belgium. [7] VIB Center for Medical Biotechnology, Ghent 9000, Belgium. [8] VIB Proteomics Core, Ghent 9000, Belgium. [9] Biocenter Oulu, University of Oulu, Oulu 90220, Finland. [10] Laboratory of Pharmaceutical Process Analytical Technology, Department of Pharmaceutical Analysis, Faculty of Pharmaceutical Sciences, Ghent University, Ghent 9000, Belgium. [11] Department of Biochemistry and Cell Biology, Faculty of Veterinary medicine, Utrecht University, Utrecht, CM 3584, The Netherlands. [12] Laboratory of Lipid Metabolism and Cancer, Department of Oncology, University of Leuven, Leuven 3000, Belgium. [13] Laboratory of Experimental Clinical Chemistry, Amsterdam University Medical Center, Amsterdam University, Amsterdam, AZ 1105, The Netherlands. [14] Biomedical Engineering & Physics, Amsterdam University Medical Center, Amsterdam University, Amsterdam, AZ 1105, The Netherlands. [15] Vesicle Observation Center, Amsterdam University Medical Center, Amsterdam, AZ 1105, The Netherlands. [16] Center of Medical Genetics, Ghent University, Ghent 9000, Belgium. Correspondence and requests for materials should be addressed to A.H. (email: An.Hendrix@ugent.be)

Biofluids contain extracellular vesicles (EV) that differ in their biogenesis, molecular patterns and cellular origin[1]. EV are involved in the pathogenesis of multiple diseases including cancer, neurodegenerative- and autoimmune- diseases, which has made EV an intensive field of research, especially for disease diagnosis, monitoring and treatment[2,3]. However, EV research and its biomedical applications are hampered by the myriad of separation methods and measurement instruments and the lack of appropriate reference materials for accurate calibration, normalization and method development[4–6]. A reference material suitable for all those purposes should (1) have EV-like physical and biochemical characteristics, hence should be from biological origin; (2) be trackable and thus be distinguishable from sample EV and (3) behave similarly as sample EV under various experimental conditions[7]. Current reference materials for calibration are polystyrene beads, silica beads or liposomes, which lack EV-like properties such as surface markers and a mean refractive index (RI) of ~1.39, resulting in inaccurate measurements[7–9]. Biological reference materials derived from biofluids or cell cultures are commercially available but they lack specificity, rendering them less suitable for quality control applications. Reference materials for normalization and method assessment are not available. Recent approaches have tried to improve the EV-like properties of existing reference materials making them more suitable for restricted flow cytometry-based use[10–12].

To meet the need for appropriate reference materials, we have generated recombinant EV (rEV) as a trackable biological reference material resembling the physical and biochemical characteristics of sample EV. rEV find their origin in the major structural component of HIV-1 virus particles, the gag polyprotein. HIV-1 gag hijacks the ESCRT (endosomal sorting complex required for transport) pathway, responsible for the release of EV, and inserts itself in membrane areas with similar lipid/protein characteristics as EV budding areas[13,14]. Expression of the polyprotein gag alone suffices for the production of nanometer-sized immature virus like particles (VLP) surrounded by a lipid bilayer and enriched for gag molecules and EV-associated proteins[14,15]. The suitability of HIV-1 VLP as a reference material for EV research has not been previously considered or tested. Here we define the physical and biochemical characteristics, trackability, stability and commutability of rEV, identify and test suitable read-out methods, provide tools to segregate rEV from sample EV for further downstream approaches and demonstrate the usability of rEV in various applications.

## Results

**rEV bear EV-like physical and biochemical traits**. rEV were produced in a well-characterized HEK293T cell culture model[16] by transient transfection with retroviral gag polyprotein C-terminally fused to EGFP (enhanced green fluorescent protein)[17]. rEV were separated from the conditioned medium (CM) 72 h after transfection of approximately $3 \times 10^9$ HEK293T cells by OptiPrep density gradient (ODG) centrifugation and consecutive pelleting resulting in ~$5 \times 10^{11}$ rEV (Fig. 1a–c). Western blot and flow cytometry analysis for gag-EGFP protein expression and viability assays of HEK293T cells revealed that the transfections were reproducible and non-toxic (supplementary fig. 1a-e). Transfection with gag-EGFP resulted in a 4.5-fold increase in EV secreted per cell compared to mock transfection (supplementary fig. 1f).

We evaluated at least three biological replicates of rEV (indicated by n in the figure legends) for their physical and biochemical characteristics that are principal to EV analysis and compared them to sample EV derived from various sources. The implementation of ODG centrifugation to separate rEV from medium conditioned by gag-EGFP transfected HEK293T cells

and to separate sample EV from urine, plasma and medium conditioned by breast cancer cells (MCF7 and 4T1) and cancer-associated fibroblasts (CAF) revealed that the buoyant density of rEV was equivalent to the buoyant density of sample EV, namely 1.086–1.119 g/mL (supplementary fig. 2). rEV and sample EV from urine, plasma or medium conditioned by MCF7, 4T1, CAF and mock transfected HEK293T cells did not significantly differ on the 0.01 significance level in size distribution (108.6 ± 9.5 nm) ($p > 0.0361$, Mann–Whitney test) and zeta potential ($-32.3 \pm 1.6$ mV) ($p > 0.0357$, Mann–Whitney test) (Fig. 2a, b). The RI of rEV, calculated by Mie theory, was ~1.37, corresponding to the RI of sample EV from various sources (Fig. 2c) (supplementary fig. 3a). Transmission electron microscopy (TEM) revealed that rEV have a cup-like morphology characteristic of sample EV (Fig. 2d).

As sample EV, rEV contain luminal and membrane-associated proteins including ALIX, tumour susceptibility gene 101 (TSG101), flotillin-1, syntenin-1, CD81 and CD9 as assessed by western blot analysis and mass spectrometry-based proteomics; tetraspanin CD63 was identified by immune-electron microscopy (Fig. 2e, f) (supplementary fig. 4). Volcano plot analysis and differential protein analysis of mass spectrometry-based proteome data further indicated that 792 out of a total of 890 proteins (89%) were equally abundant in rEV and mock EV separated by ODG centrifugation from medium conditioned by mock transfected HEK293T cells (Spearman r = 0.7712, $p < 0.0001$), including ESCRT and other EV-associated proteins (supplementary fig. 4a-d). In addition, unsupervised hierarchical clustering confirmed the technical and biological reproducibility of rEV production (supplementary fig. 4c). Mass spectrometry-based lipidomics identified common EV-associated lipid classes (including phospholipids, sphingomyelins and lysophospholipids) with a positive correlation between rEV and mock EV (Spearman r = 0.921, $p < 0.0001$) (supplementary fig. 5a). The cholesterol concentration per rEV particle did not significantly differ on the 0.01 significance level compared to cholesterol concentrations of sample EV from various sources ($p > 0.0256$, Mann–Whitney test) (supplementary fig. 5b).

**Fluorescence-, RNA- and protein-based detection of rEV**. We evaluated fluorescence-, RNA- and protein-based technologies available in routine laboratories for their suitability to quantify rEV and distinguish rEV from sample EV by exploiting features unique to rEV: fluorescence intensity, gag-EGFP protein and EGFP mRNA. We analysed at least three biological replicates of rEV (as indicated by n in the figure legends).

Fluorescence intensity was sufficient to quantify and distinguish rEV from non-fluorescent particles by fluorescent nanoparticle tracking analysis (fNTA) and fluorescence triggered high-resolution flow cytometry (HR-FC) (Fig. 3a–e). fNTA and HR-FC measured equivalent relative rEV concentrations (Fig. 3e). To establish linear or semi-logarithmic regression curves for fluorescent microplate reader, western blot, ELISA and RT-qPCR analysis, rEV were quantified by fNTA. Based on fluorescence, a microplate reader allowed to linearly deduce rEV numbers in a defined concentration range of $5 \times 10^8–1 \times 10^{10}$ rEV (Fig. 3f). At the protein level, an ELISA assay for p24, a subunit of the gag polyprotein, showed a linear correlation in a defined concentration range of $1 \times 10^6–1 \times 10^7$ rEV. Western blot analysis for gag-EGFP protein allowed to visualize rEV with a lower detection limit of $7 \times 10^7$ rEV, while densitometry analysis of the gag-EGFP protein bands revealed a linear correlation in a defined concentration range of $4 \times 10^8–1 \times 10^9$ rEV (Fig. 3g–i). At the RNA level, rEV and EGFP mRNA concentrations behaved in a semi-logarithmic manner as measured with RT-qPCR and this in a range from $1 \times 10^7$ to $1 \times 10^9$ rEV (Fig. 3j).

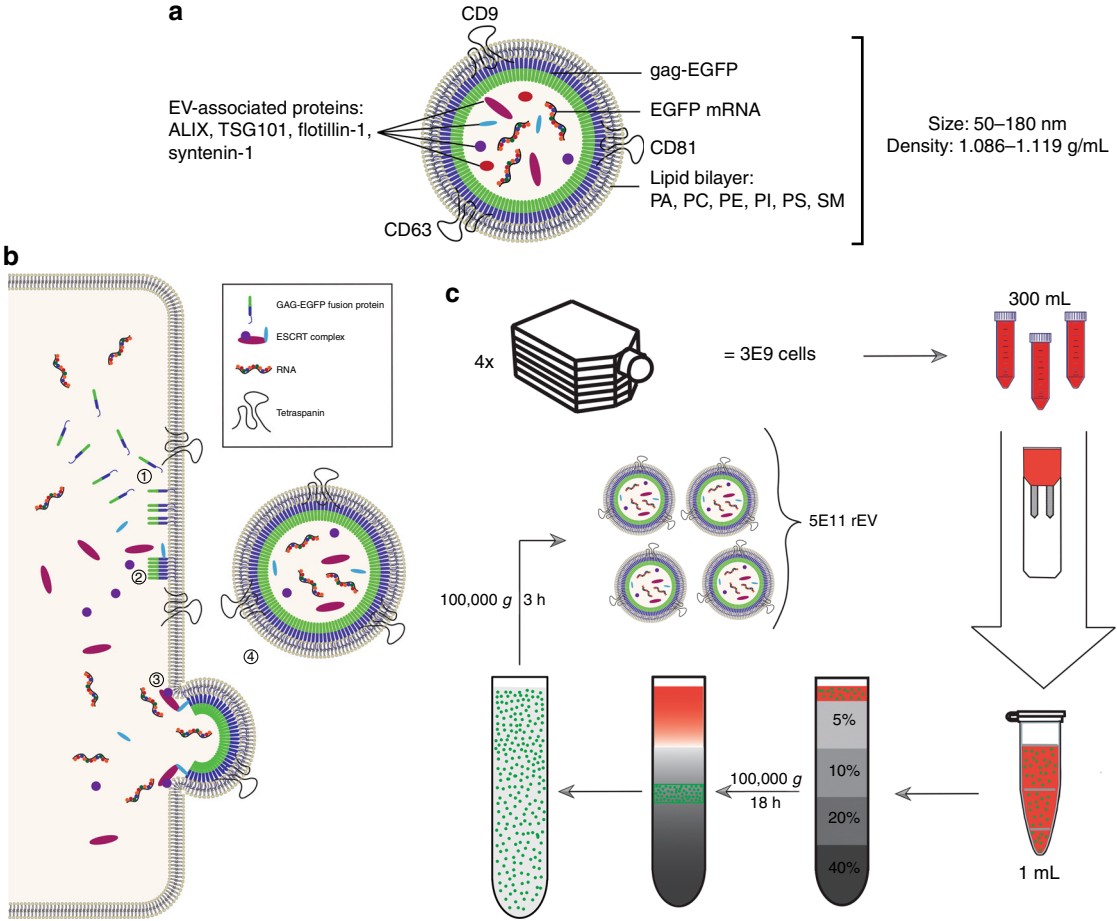

**Fig. 1** rEV are separated from conditioned medium using density gradient centrifugation. **a** Schematic representation of rEV showing representative molecular components shared with sample EV, PA phosphatidic acid, PC phosphatidylcholine, PE phosphatidylethanolamine, PI phosphatidylinositol, PS phosphatidylserine, SM sphingomyelin. **b** Schematic overview of the production of rEV at the cellular level: (1) The gag-EGFP fusion protein inserts in regions of the plasma membrane enriched for tetraspanins CD9, CD63 and CD81 via its N-terminal MA domain containing a myristoyl group. (2) The gag-EGFP fusion protein oligomerizes and recruits ESCRT-1 proteins (TSG101) via the PTAP motive on its p6 domain. (3) Recruitment of ESCRT-2/3 proteins initiates the outward budding of the gag-EGFP containing plasma membrane. (4) ESCRT-3 mediated scission of the membranes finally causes release of rEV into the conditioned medium (CM)[13]. **c** Schematic overview of the workflow to separate rEV from CM of gag-EGFP transfected HEK293T cells. Seventy-two hour post transfection CM is collected from ~3 x 10⁹ cells and concentrated to 1 mL. Concentrated CM is loaded on top of an OptiPrep density gradient (ODG) and centrifuged for 18 h at 100,000 × g. Density fractions of 1.086–1.119 g/mL are collected and pelleted for 3 h at 100,000 × g resulting in ~5 x 10¹¹ rEV per harvest

Of note, the implementation of ODG centrifugation, a density equilibrium based gradient, to separate rEV from medium conditioned by gag-EGFP transfected HEK293T cells resulted on average in 79.7% (±9.5%) fluorescent particles (Fig. 3b), indicating that gag-EGFP negative EV co-sediment with rEV. OptiPrep velocity gradient (OVG) centrifugation[18] physically segregated rEV from gag-EGFP negative EV and resulted in near 100% fluorescent particles (supplementary fig. 6a–d). Western blot analysis for gag-EGFP protein and EV-associated proteins ALIX, flotillin-1, TSG101, syntenin-1, CD81 and CD9 revealed that gag-EFGP negative EV segregate in low density fractions (1.046–1.068 g/mL, corresponding to OVG fractions 4–7) whereas rEV segregate in high density fractions (1.076–1.088 g/mL, corresponding to OVG fractions 10–13) (supplementary fig. 6a, b). rEV obtained by OVG centrifugation contained EV-associated proteins, EV-like size distribution and had a typical EV morphology as analyzed by, respectively, western blot, (f)NTA and TEM (supplementary fig. 6b–e). Mass spectrometry-based proteome data indicated that gag-EGFP negative EV segregated from rEV share 81% of the total number of detected proteins, further indicating the high similarity in protein composition of

rEV and sample EV (supplementary fig. 6j). In agreement, rEV separated by OVG centrifugation were detectable with fluorescence-, protein- and RNA-based assays (Supplementary figure 6f–h). Spiking of rEV obtained by OVG centrifugation in PBS followed by equilibrium based ODG centrifugation, identified rEV at equivalent density (1.086–1.119 g/mL) as sample EV (supplementary fig. 6i).

**rEV are structurally and biologically stable during storage**. We assessed the structural and biological stability of rEV under different storage temperatures for extended periods of time in at least three biological replicates (indicated by n in the figure legend). Freezing at −80 °C (for 6 months and 12 months) and thawing of rEV (supplementary fig. 7a, b), as well as storage of rEV at 4 °C up to 7 days (supplementary fig. 7c, d) did not affect their density, fluorescence intensity, number and size distribution, as evaluated by ODG centrifugation and fNTA (supplementary fig. 7e). Immunoprecipitation with an antibody recognizing the extravesicular loop of the tetraspanin CD81 was unaffected following a freeze-thaw cycle demonstrating the correct orientation

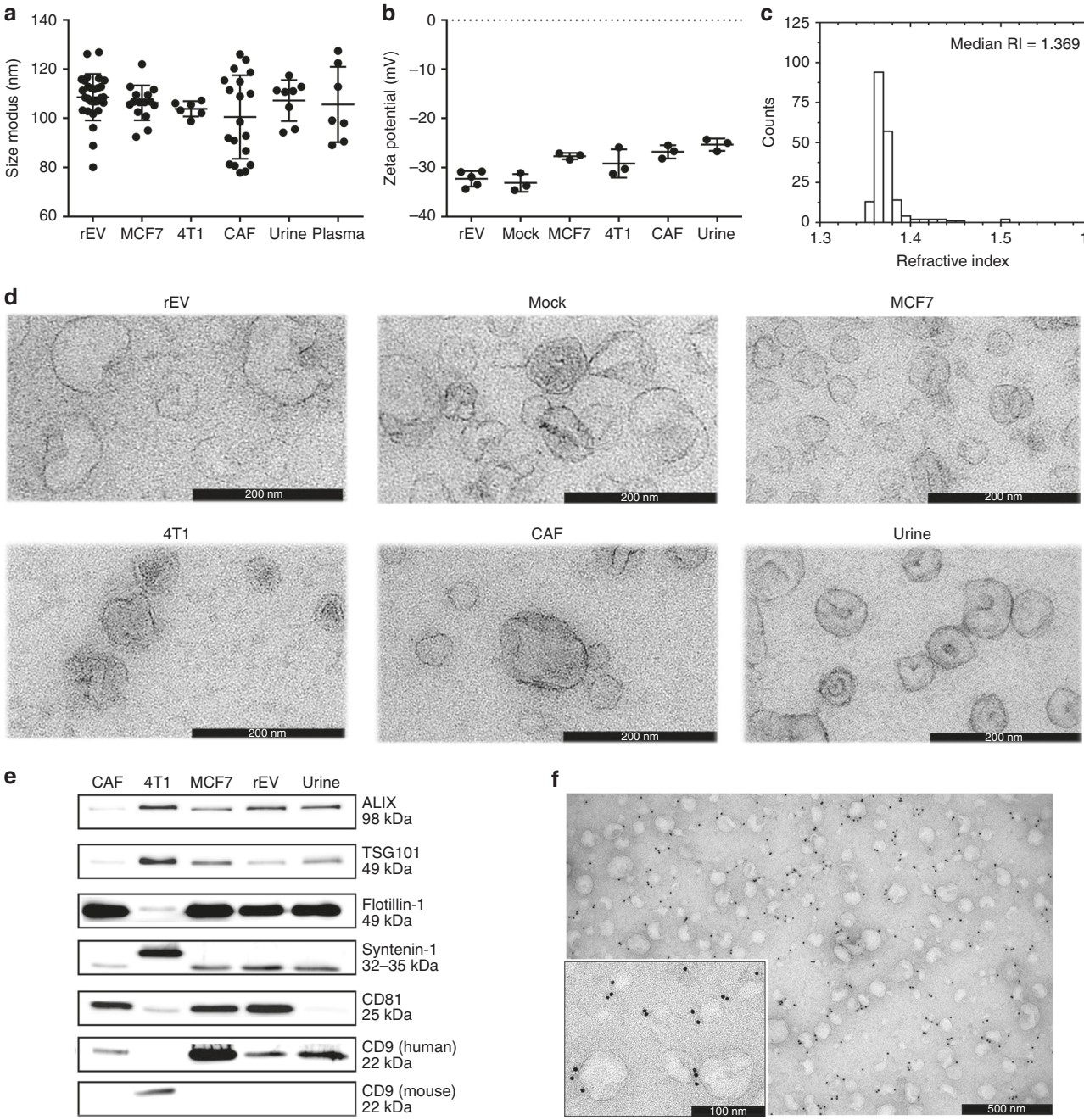

**Fig. 2** rEV bear physical and biochemical traits characteristic of sample EV. rEV, separated by ODG centrifugation of medium conditioned by gag-EGFP transfected HEK293T cells, are compared to sample EV separated by ODG centrifugation from different sources, including medium conditioned by breast cancer cells (MCF7, 4T1), mock transfected HEK293T cells (mock) or cancer-associated fibroblasts (CAF) or plasma and urine for **a** size distribution measured with nanoparticle tracking analysis (NTA) (*n* > 6), **b** zeta potential (*n* > 3), **c** refractive index distribution calculated with NTA and MIE theory (supplementary fig. 3a), **d** morphology as imaged by transmission electron microscopy (TEM) and the presence of EV-associated proteins ALIX, TSG101, flotillin-1, syntenin-1, CD81, CD9 and CD63 analysed by **e** western blot analysis (30 µg protein loaded on gel) and **f** immune-electron microscopy with a secondary gold labelled antibody against a primary antibody targeting the extracellular loop of the tetraspanin CD63. Images are representative of three biological replicates. Data in **a** and **b** are (mean, SD). Source data are provided as a source data file

of the rEV membrane (supplementary fig. 7f). Also, lyophilization of rEV in PBS supplemented with 5% trehalose did not affect their morphology, fluorescence intensity, number, size distribution and density as assessed by TEM, HR-FC, fNTA and ODG centrifugation (supplementary fig. 8). Trehalose (5%) was needed and sufficient since absence of this sugar resulted in reduced rEV numbers, increased size and membrane disruption as assessed by fNTA and TEM (supplementary fig. 9).

**rEV define recovery efficiencies of EV separation methods**. To assess whether rEV are fit to test the performance of separation methods, $5 \times 10^{10}$ rEV were spiked in PBS, non-cell exposed culture medium (DMEM), plasma and urine followed by size-, immune affinity- and density-based separation of rEV from corresponding biofluids.

Western blot analysis for gag-EGFP protein and EV-associated proteins flotillin-1 and TSG101 revealed that rEV elute in

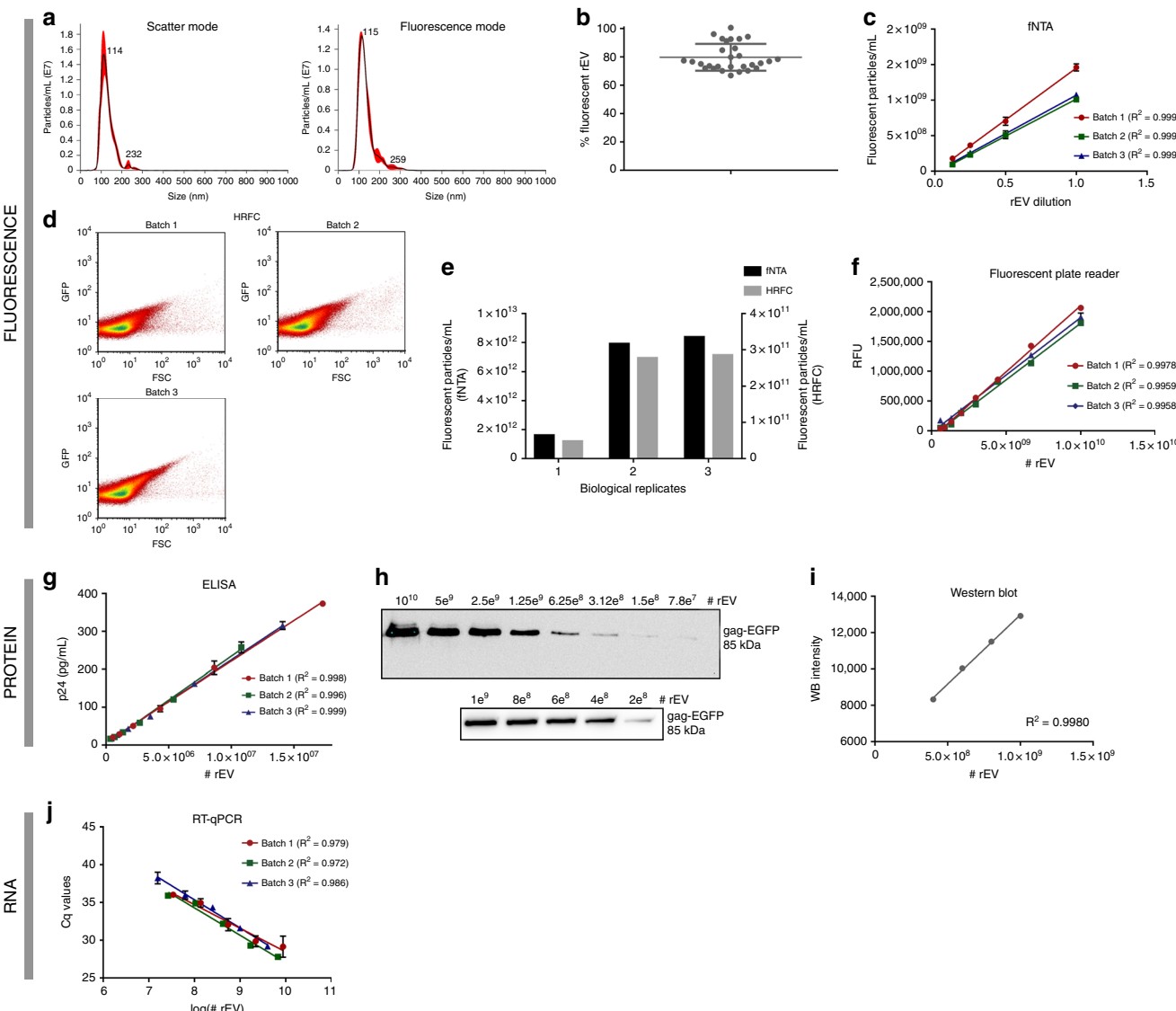

**Fig. 3** rEV can be detected and quantified using fluorescence-, protein- and RNA-based assays. rEV can be directly quantified using fluorescent NTA (fNTA) and fluorescence triggered high-resolution flow cytometry (HR-FC). **a** Size distributions under both scatter and fluorescence NTA mode and **b** the percentage of fluorescent rEV (ratio fNTA/NTA) detected above detection threshold ($n = 29$). **c** Linear correlation analysis of ½ dilutions of rEV in PBS measured by fNTA ($n = 3$). **d** Scatter plots showing quantification of rEV with HR-FC ($n = 3$). **e** Relative concentrations of rEV directly quantified by fNTA versus HR-FC ($n = 3$). rEV can be indirectly quantified using fluorescent plate reader, p24 ELISA, western blot and RT-qPCR. **f** Linear correlation analysis of relative fluorescence units (RFU) measured by a fluorescent plate reader versus number of rEV ($n = 3$). **g** Linear correlation analysis of p24 concentration measured by a p24 ELISA assay versus number of rEV ($n = 3$). **h** Western blot analysis for EGFP on the maximum detectable concentration range with **i** linear correlation analysis of protein band intensity versus rEV number. **j** Semi-logarithmic correlation analysis of EGFP mRNA Cq values calculated by RT-qPCR versus number of rEV ($n = 3$) (measurements in **c**, **f**, **g**, **j** are performed in triplicate and presented as (mean, SD)). Source data are provided as a source data file

identical 1 mL elution volume fractions (4, 5 and 6) as sample EV upon size exclusion chromatography (SEC) (Fig. 4a). Similar to sample EV from urine and plasma, western blot analysis for gag-EGFP protein and EV-associated proteins syntenin-1 and ALIX revealed that rEV co-precipitate with antibodies directed against the large extravesicular loop of CD81 or CD63 (Fig. 4b). rEV spiked in urine or culture medium floated at equivalent density (1.086–1.119 g/mL) as sample EV when separated by ODG centrifugation (Fig. 4c) (supplementary fig. 2). Thus, for all these separation methods, rEV are commutable, i.e. behave similar as sample EV when spiked in these biofluids.

Of note, rEV spiked in plasma shifted to higher densities compared to sample EV (1.141–1.215 g/mL vs 1.086–1.119 g/mL)

as shown by fNTA and western blot analysis for gag-EGFP protein (supplementary fig. 10a, b). This shift was reversible, as proteinase K (PK) treatment of the concentrated SEC elution volume fractions (4, 5 and 6) prior to ODG centrifugation reversed the density shift (Fig. 4c) (supplementary fig. 10b, c). Similar results were obtained with rEV separated by OVG centrifugation from medium conditioned by gag-EGFP transfected HEK293T cells (supplementary fig. 6i). PK treatment had no effect on fluorescence intensity and integrity of rEV nor on the number and quality of sample EV as evidenced by the presence of luminal-membrane-associated proteins syntenin-1 and flotillin-1 in PK treated plasma samples, and (f)NTA measurements (supplementary fig. 10b–e). This shift was inducible, as addition

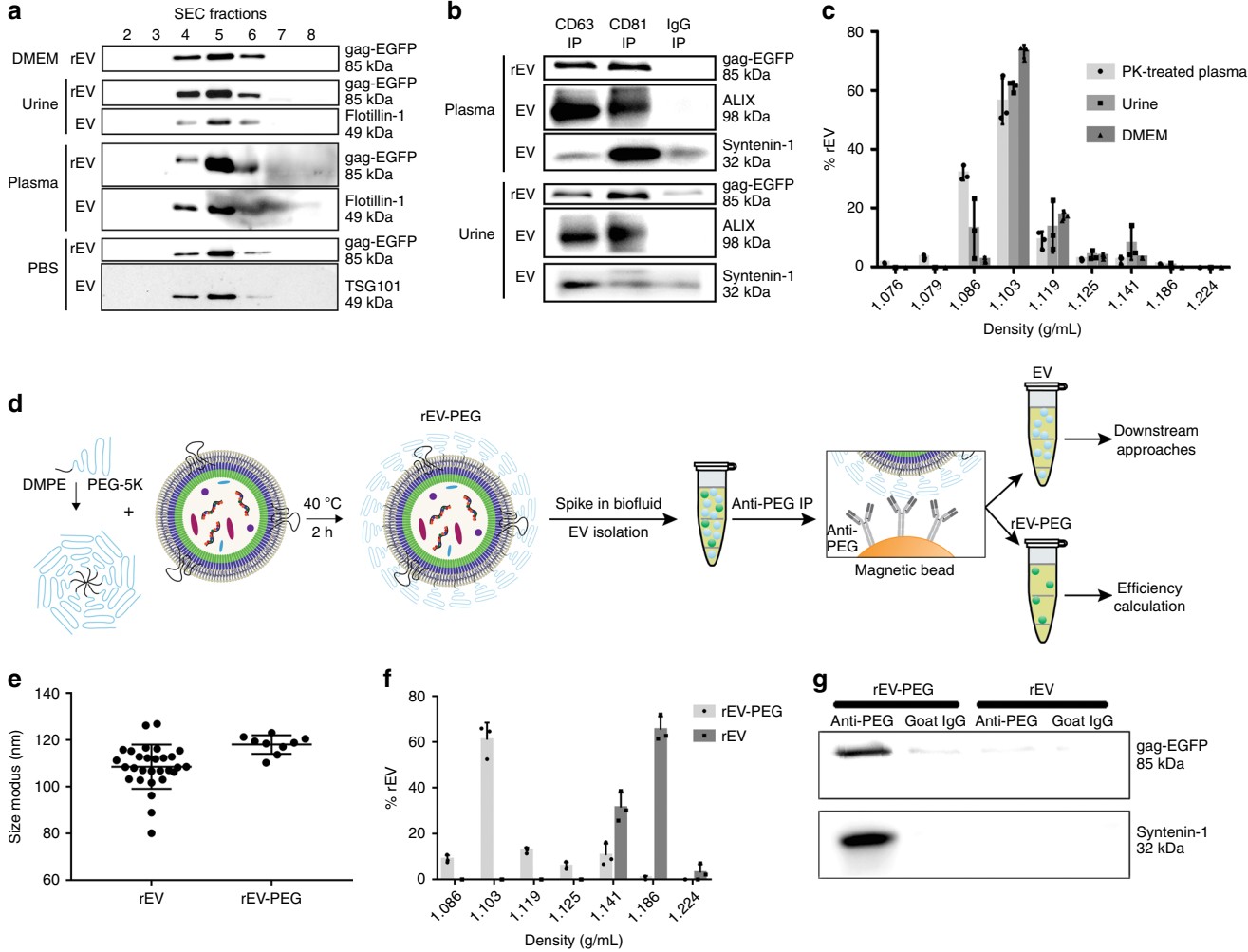

**Fig. 4** rEV and rEV-PEG act as sample EV when spiked in multiple biofluids. **a** Western blot analysis for EGFP, flotillin-1 and TSG101 of size exclusion chromatography (SEC) fractions from DMEM, urine, plasma and PBS (containing EV from MCF7 breast cancer cells) spiked or not with rEV. rEV is indicative of rEV spike; EV is indicative of sample EV (i.e. non-spiked biofluid). **b** Western blot analysis for EGFP, ALIX and syntenin-1 of plasma and urine spiked or not with rEV and immune precipitated using anti-CD81, anti-CD63 or non-specific goat IgG coated magnetic beads (IP). rEV is indicative of rEV spike; EV is indicative of sample EV (i.e. non-spiked biofluid). **c** Density distribution of rEV spiked in urine, DMEM and proteinase K (PK) treated plasma, obtained by ODG centrifugation followed by fNTA measurement of ODG fractions ($n = 3$). **d** Schematic of rEV post-insertion PEGylation to make rEV compatible with multiple biofluids and to allow segregation of rEV from sample EV prior to downstream analysis of sample EV. **e** Size distribution of rEV and rEV-PEG measured by fNTA ($n > 9$). **f** Density distribution of rEV-PEG and rEV spiked in plasma, obtained by ODG centrifugation followed by fNTA measurement of ODG fractions ($n = 3$). **g** Western blot analysis for EGFP and syntenin-1 of immune precipitated rEV and rEV-PEG in PBS with anti-PEG or non-specific goat IgG coated magnetic beads (image representative of two independent experiments). Data in **c**, **e**, **f** are presented as (mean, SD). Source data are provided as a source data file

of physiological amounts of IgM (100 μg) or IgG (5 mg) induced the density shift of rEV spiked in PBS and urine (supplementary fig. 10f).

Although this density shift is a unique feature to segregate rEV from sample EV, it is only applicable using density-based separation methods in plasma or IgG/IgM supplemented biofluids. To enable rEV segregation from sample EV in any biofluid using any separation method, we modified rEV by post-insertion PEGylation to avoid protein corona formation (Fig. 4d). Surface masking of rEV by DMPE-PEG (1,2-dimyristoyl-sn-glycero-3-phosphoethanolamine conjugated to polyethyleneglycol) increased the mean size distribution by 10 nm ($p = 0.0061$, Mann–Whitney test) but did not alter fluorescence intensity, prevented the interaction with corona proteins and provided the unique opportunity for proteinase-independent segregation of spiked rEV from sample EV. More than 85% of rEV-PEG spiked

in plasma floated at similar densities as sample EV (1.086–1.119 g/mL) (Fig. 4e, f). Additionally, immune precipitation using anti-PEG-coated magnetic beads specifically captured rEV-PEG and not rEV (Fig. 4g).

**rEV mitigate for intra-method and inter-user variability.** The efficiency of EV recovery, defined as the number of rEV detected after separation divided by the number of rEV spiked ($5 \times 10^{10}$) (expressed as a percentage), of frequently used separation methods was calculated by rEV quantification with fNTA and anti-p24 ELISA. Efficiencies of SEC[19], ODG centrifugation[20] and differential ultracentrifugation (dUC)[21] to separate rEV from plasma were, respectively, nearly 100, 30 and 10%. Recovery efficiency of rEV from ODG density fractions 1.086–1.119 g/mL by a $100,000 \times g$ pelleting step instead of SEC was reduced from 30% to only 5%

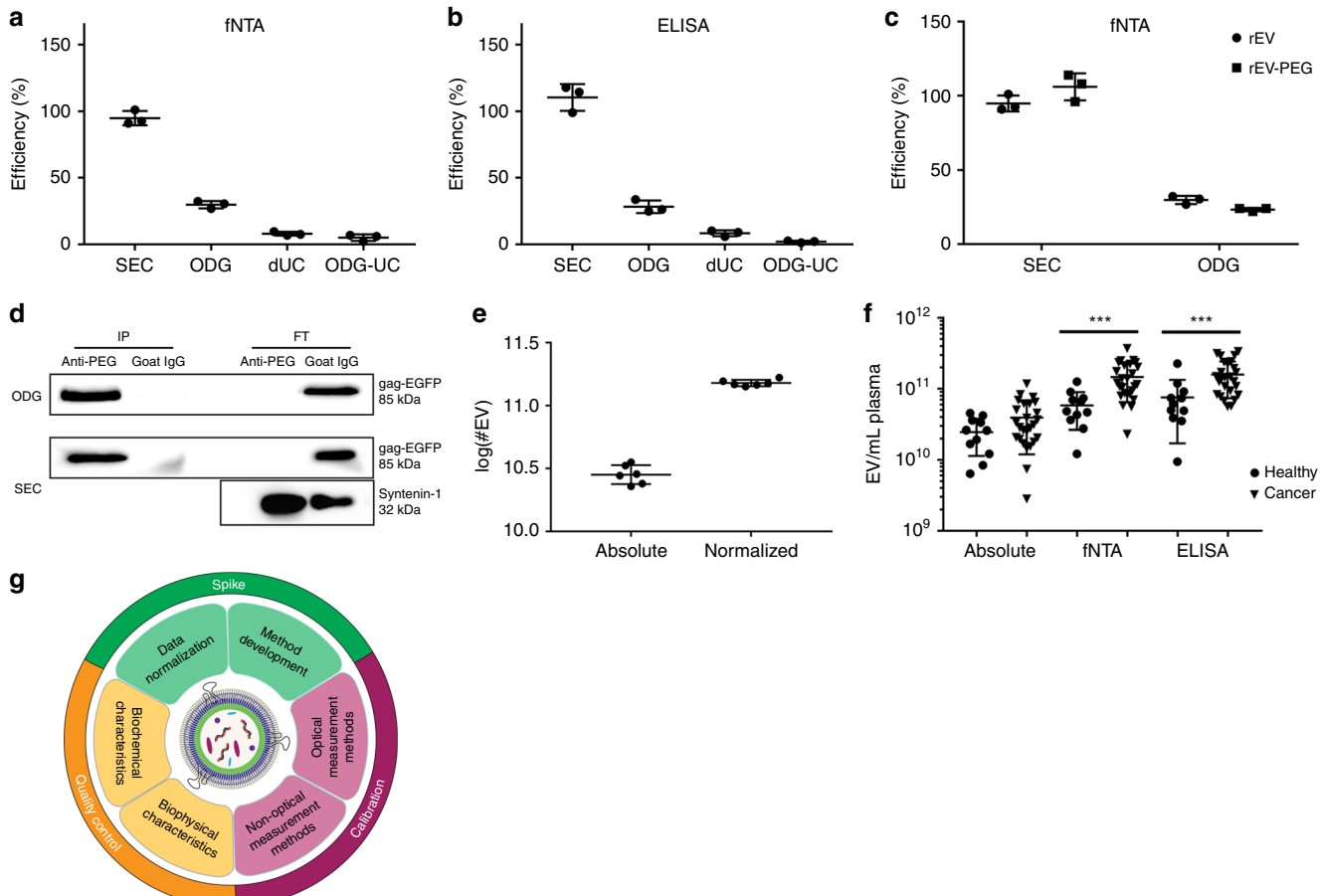

**Fig. 5** Use of rEV in various applications. **a** Calculation of efficiencies of EV separation methods from plasma by measuring spiked rEV ($5 \times 10^{10}$) with fNTA and **b** an ELISA for p24, a subunit of the gag polyprotein ($n = 3$). **c** Comparison of calculated EV separation efficiencies for plasma by fNTA, making use of rEV or rEV-PEG ($n = 3$). **d** Western blot analysis for EGFP and syntenin-1 of immune precipitated rEV-PEG and sample EV after separation from plasma by SEC or ODG centrifugation, making use of anti-PEG or non-specific goat IgG coated magnetic beads (IP: immune precipitated, FT: flow through) (image representative of two biological replicates). **e** Log of the number of sample EV measured by NTA before and after normalization by rEV quantification with fNTA to reduce inter-user variability (graph representative of two biological replicates). **f** EV/mL plasma from healthy volunteers ($n = 11$) and breast cancer patients ($n = 26$) after separation by ODG centrifugation and normalization by rEV quantification with fNTA and an ELISA for p24. ***$P < 0.001$ (Mann–Whitney test). **g** Schematic overview of various rEV applications. All data depicted in **a**, **b**, **c**, **e**, **f** are (mean, SD). Source data are provided as a source data file and supplementary tables 1 and 2

(Fig. 5a, b). The number of rEV spiked ($1 \times 10^8$–$1 \times 10^{11}$) did not impact recovery calculations (supplementary fig. 11). In addition, recovery calculations were independent of the use of rEV or rEV-PEG (Fig. 5c). After separation of rEV-PEG and sample EV from plasma by SEC and ODG centrifugation, the use of anti-PEG antibody-coated magnetic beads allowed for a complete segregation of rEV-PEG from sample EV (Fig. 5d).

The coefficient of variation (CV), a measure for method repeatability defined by the SD divided by the mean (expressed as a percentage), is a pertinent statistic for accurate data interpretation of EV separation and quantification methods. In general, intra-method CV should be less than 10%. Using rEV, the intra-method CV for the EV separation methods SEC, ODG centrifugation and dUC could be calculated and were, respectively, 5.7%, 9.2% and 18.0% as measured by fNTA ($n = 3$) (Fig. 5a). The intra-method CV for rEV quantification methods fNTA, RT-qPCR and anti-p24 ELISA were, respectively, 5.8%, 0.8% and 5.7% (Fig. 3c, g, j). To demonstrate the applicability of rEV to define and mitigate for inter-user variations, $1 \times 10^{10}$ rEV were spiked in plasma and separated by ODG centrifugation ($n = 6$). Inter-user variation was induced by replacing defined sample volumes with PBS (see material and methods). The total

number of sample EV and rEV were quantified by NTA and fNTA, respectively. The inter-user variation, expressed as CV, showed a 66% reduction (from 17.5% to 5.9%) due to normalization for rEV recovery efficiencies (Fig. 5e) (supplementary table 1).

**rEV accurately enumerate EV in patient samples**. Sample EV and spiked rEV ($2 \times 10^{10}$) were separated by ODG centrifugation from 2 mL of plasma of breast cancer patients ($n = 26$) and sex-matched healthy volunteers ($n = 11$), and were measured by NTA and fNTA, respectively. The average, normalized sample EV concentration per mL plasma was increased by 2.5 fold ($1.46 \times 10^{11}$ vs $5.83 \times 10^{10}$) ($p = 0.0002$, Mann–Whitney test) in breast cancer patients compared to healthy individuals. In contrast, absolute counts did not reveal a significant difference ($3.91 \times 10^{10}$ vs $2.47 \times 10^{10}$) ($p = 0.1410$, Mann–Whitney test). Normalization was independent from the rEV quantification method used since implementation of a p24 ELISA to quantify rEV provided equivalent results in breast cancer patients compared to healthy individuals ($1.59 \times 10^{11}$ vs $7.55 \times 10^{10}$) ($p = 0.0009$, Mann–Whitney test) (Fig. 5f) (supplementary table 2).

## Discussion

The need for a biological reference material for EV research and biomedical applications is increasingly recognized[5,9,22]. To meet this need, we have developed rEV, a gag polyprotein-induced mimic of EV. rEV were generated by reproducible transfection of HEK293T cells with gag-EGFP DNA followed by high-purity rEV separation from CM using density gradient centrifugation[23]. rEV are safe, since the viral genome, proteases and other viral proteins, which render viruses infectious are not present[24]. rEV show biochemical and physical traits characteristic of sample EV[14,15,25–27]: (1) equivalent lipid and protein profiles including the presence of intraluminal and membrane EV-associated proteins; and (2) equivalent cup-shaped morphology, size (50–180 nm), buoyant density (1.086–1.119 g/mL), RI (1.37) and zeta potential (−32 mV). rEV are stable during lyophilization and long-term storage, ensuring convenient distribution and use. They contain exogenous mRNA and protein allowing quantification with multiple measurement instruments while ensuring adequate differentiation from sample EV. rEV are the first biological reference material for EV compatible with a plethora of applications including quality control, instrument calibration and data normalization (Fig. 5g).

Mass spectrometry-based proteomics revealed that rEV contain intraluminal and membrane proteins characteristic of sample EV including ALIX, TSG101, flotillin-1, syntenin-1, CD9, CD81 and CD63. In depth analysis for molecular function and biological processes revealed that rEV are enriched in proteins regulating RNA binding and nucleic acid metabolism (supplementary fig. 4e). Gag polyprotein contains zinc-finger RNA binding domains and to form VLP, gag polyprotein must bind RNA[28,29]. In the absence of viral RNA, gag encapsulates host RNA and any single-stranded nucleic acid longer than ~20–30 nt can support VLP assembly, indicating a general propensity to bind abundant RNA[30,31]. Indeed, exogenous EGFP mRNA is encapsulated in rEV and allows for a reproducible quantification of rEV using EGFP or gag RT-qPCR assays. Fusion of EGFP to gag polyprotein results in the delivery of exogenous gag-EGFP protein to rEV allowing for a reproducible quantification of rEV using fluorescence-based fNTA, HR-FC and plate reader or protein-based p24 ELISA assays. Inherently, the number of spiked rEV appropriate for data normalization depends on the recovery efficiency of the separation method and the sensitivity of the rEV detection method. One or more of these detection methods are available in any routine laboratory allowing for a broad implementation of rEV. Thus, rEV can be differentiated from sample EV, including sample EV derived from body fluids of HIV patients, due to the presence of exogenous EGFP mRNA and gag-EGFP protein. rEV are highly adaptable and can be used to deliver RNA and proteins as part of gag or not, alike EGFP mRNA and protein, opening perspectives for custom-made rEV for multiple diagnostic and therapeutic applications[32].

The separation of rEV by ODG centrifugation from medium conditioned by gag-EGFP transfected HEK293T cells resulted in ~80% fluorescent particles, implying that not all particles contain gag-EGFP protein. Indeed, separation of rEV by OVG centrifugation resulted in ~100% fluorescent particles, indicating that EV generated by endogenous biogenesis of HEK293T cells are segregated from rEV. In comparison, other proposed fluorescent reference EV such as EV containing a membrane-associated form of GFP (palm-GFP)[33] or lipophilic fluorophore stained EV (Hansabiomed Life Sciences) delivered less than 13% fluorescent particles as measured by fNTA (supplementary fig. 12). Also, only 16% of EV containing tetraspanin-EGFP fusion proteins, recently proposed and validated as biological reference material for restricted use in flow cytometry platforms, were detectable by fNTA analysis[12].

rEV have a RI, a physical property determining the amount of light that is scattered by a certain material, equivalent to sample EV, indicating that rEV can be used as a calibrator for measurement instruments. To compare or reproduce EV research, exact enumeration of EV particles rather than indirect quantification via protein concentration measurements is recommended[5]. Flow cytometry and NTA are frequently used to quantify individual particles. Synthetic silica or polystyrene beads with a RI of respectively 1.45 and 1.61 are commonly used to calibrate these instruments. Since the RI determines the minimal size of particles detected with NTA and determines the relationship between scatter and size in flow cytometry, the average size and concentration of sample EV detected by these methods is underestimated[7,8]. Recent advances in the field of metrology have possibly bypassed this problem with the standardized production of hollow organosilica beads (HOB)[10]. Although HOB have an equivalent RI to sample EV, in contrast to rEV they do not contain EV-enriched surface markers, limiting their applicability to light scattering.

To be fit for their intended use, rEV must be commutable, meaning that they must perform equally to sample EV undergoing the actual procedure[34,35]. We have demonstrated that rEV, modified or not by post-insertion PEGylation, act as sample EV when spiked in multiple biofluids. This allowed us to estimate the recovery efficiency of sample EV from plasma using commonly implemented separation methods including dUC, SEC and ODG centrifugation[4]. The recovery efficiency of dUC varied between 5 and 10%. This finding, together with previous reports showing that dUC co-pellets contaminants and disrupts sample EV[23,36] further advices against the use of dUC to retrieve EV from plasma. SEC recovered rEV with 100% efficiency but, as previously reported, is unable to appropriately resolve sample EV from low density lipoproteins, chylomicrons and protein aggregates[20,37,38]. The use of ODG centrifugation to separate rEV from plasma resulted in 30% recovery efficiency. Nevertheless, the orthogonal implementation of size and density-based separation of sample EV has been reported to separate EV with very high specificity[20,37]. These examples illustrate that rEV can assist in identifying the performance of EV separation methods. In addition, in a clinical set-up rEV mitigated for intra-assay (manually prepared SEC columns, products from different batches) and inter-user (different time points, different sample handling) variability while separating EV from plasma samples of healthy donors and cancer patients resulting in increased sensitivity in EV enumerations. Indeed, recovery efficiencies of rEV varied between time points and donors (supplementary table 2).

After separation and recovery efficiency calculation, post-insertion PEGylation of rEV allows for segregation of rEV-PEG from sample EV using anti-PEG antibodies. As such, rEV-PEG allows for normalization during separation but after segregation, they will not interfere with downstream analysis of sample EV composition (proteomics, lipidomics, transcriptomics and metabolomics) and function. rEV-PEG are compatible with EV separation methods that do not require immune capture by targeting EV surface proteins since antibody binding is sterically hindered by PEG.

Interlaboratory evaluation and benchmarking of rEV to assess the applicability for data normalization, instrument calibration and quality control will be of great importance to stimulate their wide-spread use. Whether rEV, with a size distribution between 50 and 180 nm, are appropriate as biological reference material for large EV (>200 nm) remains to be investigated and is likely dependent upon the separation and measurement method of choice (use of size and large EV-specific protein versus other isolation or characterization methods).

In conclusion, we propose rEV as a biological reference material to be implemented in EV separation methods and

measurement instruments. rEV have equivalent buoyant density, size distribution, morphology, RI, zeta potential and molecular patterns (proteins and lipids) to sample EV. The gag-EGFP fusion protein enables sensitive and differential trackability of rEV. As such rEV are tailor-made for quality control, data normalization, method development and accurate calibration of optical and non-optical EV measurement methods (Fig. 5g). The use of rEV will ensure rigorous sample EV analysis and is essential to advance the field and develop future EV-based biomedical applications.

## Methods

**Antibodies**. The following primary and secondary antibodies were used for immunostaining: mouse monoclonal anti-ALIX (1:1000, #2171)and rabbit monoclonal anti-CD9 clone D3H4P (1:1000, #13403S) (Cell Signaling Technology, Danvers, MA, USA), mouse monoclonal anti-CD63 clone MEM-259 (1:200, #ab8219) and rabbit monoclonal anti-syntenin-1 (1:1000, #ab133267) Abcam, Cambridge, UK), mouse monoclonal anti-CD81 (1:1000, #SC-166029) and mouse monoclonal anti-TSG101 (1:100, #SC-7964) (Santa Cruz Biotechnology, Dallas, TX, USA), mouse monoclonal anti-flotillin-1 (1:1000, #610820) and rat anti-mouse CD9 (1:1000, #553758) (BD Biosciences, Franklin Lakes, NJ, USA), mouse monoclonal anti-green fluorescent protein (GFP) (1:1000, #MAB3580) (Merck Millipore, Billerica, MA, USA), mouse monoclonal anti-α-tubulin (1:4000) (T5168, Sigma, Diegem, Belgium), sheep anti-mouse horseradish peroxidase-linked antibody (1:3000, #NA931V) and donkey anti-rabbit horseradish peroxidase-linked antibody (1:4000, #NA934V) (GE Healthcare Life Sciences, Uppsala, Sweden). Immune-electron microscopy was performed with a primary mouse monoclonal anti-CD63 antibody (1:50) (clone H5C6) (557305, BD Biosciences, Franklin Lakes, NJ, USA) and a rabbit anti-mouse IgG (1:2000) (Zymed Laboratories, San Francisco, CA, USA) to which 10 nm Gold-conjugated Protein A was added (1:70) (Cell Microscopy Core, University Medical Center Utrecht, The Netherlands). Antibodies used for immunoprecipitation were mouse monoclonal anti-CD81 antibody (MA5-13548, Thermo Fischer scientific, Erembodegem, Belgium), mouse monoclonal anti-CD63 (556019, BD Biosciences, Franklin Lakes, NJ, USA) and rabbit polyclonal anti-PEG (ab190652, Abcam, Cambridge, UK).

**Cell culture and transfection**. Human HEK293T[16] and MCF7 cells, mouse 4T1 cells (ATCC, Manassas, VA, USA) and human cancer-associated fibroblasts[39] were cultured in a humidified atmosphere at 10% (4T1 at 5%) $CO_2$ using high glucose DMEM (Invitrogen, Carlsbad, CA, USA) supplemented with 10% fetal bovine serum (Greiner Bio One, Kremsmünster, Austria), 100 U/mL penicillin and 100 μg/mL streptomycin (Life Technologies, Carlsbad, CA, USA). Cells were passaged at 70–80% confluency in T175 flasks (Greiner Bio One, Kremsmünster, Austria) and were discarded after 10 passages. Cell cultures were regularly tested and confirmed negative for mycoplasma contamination using the MycoAlert Mycoplasma Detection Kit (Lonza, Verviers, Belgium).

pMET7-gag-EGFP and empty pMET7mcs (mock) plasmids, were purified from DH10b E. coli using the PC2000 nucleobond kit (Macherey-Nagel, Düren, Germany) following the manufacturer's procedures[17]. The same gag-EGFP plasmid can be purchased at addgene (# 80605). The pMET7mcs plasmid was obtained by insertion of a multicloning site (GAATTCTAATACGACTCACTATAGGGAGTC GACTCAGATCTTCGATATCTCGGTAACCTCACCGGTTCCTCGAGTCTCTA GA) in the EcoRI-XbaI site of the pME18S-FL3-3 vector (Genbank:AB009864.2). To produce rEV, HEK293T cells were seeded in Falcon cell culture Multi-Flasks (Corning, New York, USA) and transiently transfected at 70–80% confluency using 25 kDa linear polyethyleneimine (PEI) (Polysciences, Warrington, PA, USA) in a PEI:DNA ratio of 5:1 with a final concentration of 1 μg DNA/mL culture medium in a total volume of 120 mL per Multi-Flask.

**rEV separation from cell culture medium**. 48 hours following transfection, cells were washed three times using Opti-MEM (31985070, Thermo Fischer Scientific, Erembodegem, Belgium) followed by 24 h incubation with 75 mL Opti-MEM supplemented with 100 U/mL penicillin and 100 μg/mL streptomycin (Life Technologies, Carlsbad, CA, USA) at 37 °C and 10% $CO_2$. Conditioned medium (CM) was harvested and centrifuged for 10 min at $200 \times g$ and 4 °C to remove detached cells, followed by a 0.45 μm cellulose acetate filtration (Corning, New York, USA) to remove larger particles. Next, CM was concentrated ~300 times using a Centricon Plus-70 centrifugal filter device with a 10 kDa nominal molecular weight limit (Millipore, Burlington, MA, USA). The resulting concentrated CM (CCM) was filtered through a 0.2 μm cellulose acetate filter (GE Healthcare Life Sciences, Uppsala, Sweden) and 1 mL was used for OptiPrep density gradient (ODG) ultracentrifugation. Following collection of the medium, cell cultures were trypsinized and cell viability was measured on a Countess Automatic Cell Counter (Thermo Fischer Scientific, Erembodegem, Belgium) using a 0.1% trypan blue exclusion test included in the kit.

A discontinuous ODG was made by layering 4 mL of 40%, 4 mL of 20%, 4 mL of 10% and 3.5 mL of 5% iodixanol solutions (Axis-Shield, Oslo, Norway) on top of each other in a 16.8 mL open top polyallomer tube (337986, Beckman Coulter, Brea,

CA, USA)[23]. One millilitre CCM sample was overlaid on top of the gradient that was then centrifuged for 18 h at $100,000 \times g$ and 4 °C (SW 32.1 Ti rotor, Beckman Coulter, Brea, CA, USA). All gradients were made using a biomek 4000 automated workstation (Beckman Coulter, Brea, CA, USA). Solutions of 5, 10, 20 and 40% iodixanol were made by mixing appropriate amounts of a homogenization buffer (0.25 M sucrose, 1 mM EDTA, 10 mM Tris-HCL, [pH 7.4]) and a 50% iodixanol working solution. This working solution was prepared by combining a working solution buffer (0.25 M sucrose, 6 mM EDTA, 60 mM Tris-HCl, [pH 7.4]) and a stock solution of OptiPrep (60% (w/v) aqueous iodixanol solution) (Axis-Shield, Oslo, Norway). After centrifugation, gradient fractions of 1 mL were collected from top to bottom using the biomek 4000 automated workstation, fractions 8, 9 and 10, corresponding to a buoyant density of 1.086–1.119 g/mL, were collected, pooled and diluted to 16 mL in PBS and centrifuged for 3 h at $100,000 \times g$ and 4 °C (SW 32.1 Ti rotor, Beckman Coulter, Brea, CA, USA). The resulting pellet was resuspended in 100 μL PBS and stored as 10 μL aliquots at −80 °C after which they could be thawed by holding the tube by hand and monitoring thawing. For proteomics and lipidomics the last $100,000 \times g$ pelleting step was replaced by size exclusion chromatography (SEC) (see further)[40]. To estimate the density of each fraction a standard curve was made of the absorbance values at 340 nm of 1:1 aqueous dilutions of 5, 10, 20 and 40% iodixanol solutions. This standard curve was used to determine the density of fractions collected from a control gradient overlaid with 1 mL of PBS.

**EV separation from cell culture medium**. For the isolation of EV from MCF7, 4T1 and cancer-associated fibroblast cells, the same procedure as for the isolation of rEV was used except for the washing of the cells. These cell lines were washed and incubated for 24 h with DMEM supplemented with 0.5% EV-depleted fetal bovine serum (EVDS). EVDS was obtained through 18 h centrifugation of fetal bovine serum at $100,000 \times g$ and subsequent filtering through a 0.2 μm filter (GE Healthcare Life Sciences, Uppsala, Sweden).

**Sample collection**. Collection of patient samples was according to ethical committee of Ghent University Hospital approval and in accordance to relevant guidelines. Venous blood from sex-matched healthy volunteers and early breast cancer patients was collected using Venosafe-citrate tubes (VF-054SBCS07, Terumo Europe, Leuven, Belgium). Directly after collection, whole blood was centrifuged two times for 15 min at $2500 \times g$ at room temperature resulting in platelet-depleted plasma (PDP), as was verified by a negative platelet count ($0 \times 10^4$ plt/μL) measured with an hemato analyzer. PDP was stored at −80 °C until further use.

Urine from healthy volunteers was collected and centrifuged for 10 min at $1000 \times g$ and 4 °C followed by direct EV isolation.

Only if stated otherwise, all samples were obtained from healthy volunteers.

**EV separation methods for plasma**. Differential ultracentrifugation (dUC) was performed according to Thery et al.[21]. In short, 6 mL plasma was diluted 1:1 in PBS and centrifuged at $2000 \times g$ for 30 min at 4 °C, the supernatant was then centrifuged once at $12,000 \times g$ for 45 min and 2 h at $110,000 \times g$ at 4 °C in a SW32.1 Ti rotor (Beckman Coulter, Brea, CA, USA). The resulting pellet was then diluted to 5 mL PBS, 0.22 μm filtered and centrifuged again at $110,000 \times g$ for 70 min in a SW55 Ti rotor (Beckman Coulter, Brea, CA, USA). The final pellet was resuspended in 100 μL PBS and analysed directly with fNTA (see further) or stored at −80 °C for ELISA measurement.

Size exclusion chromatography (SEC) was performed in a 10 mL syringe (BD Biosciences, Franklin Lakes, NJ, USA) with a nylon net with 20 μm pore size (NY2002500, Merck Millipore, Billerica MA, USA) at the bottom. The syringe was packed with 10 mL pre-washed Sepharose CL-2B (GE Healthcare, Uppsala, Sweden) and 2 mL plasma was loaded on top, after which 1 mL fractions of eluate were collected under constant gravitational flow by continuously adding PBS containing 0.32% trisodiumcitrate dihydrate (ChemCruz, Dallas, Texas, USA). The collected fractions were analysed directly with NTA or frozen at −80 °C for protein analysis.

To further purify EV and/or rEV from plasma, following SEC, eluted fractions 4–5–6 were pooled and concentrated to 1 mL with Amicon Ultra-2mL centrifugal filters with a 10 kDa cut-off value (UFC201024, Merck Milipore, Billerica MA, USA) and placed on top of a discontinuous ODG and centrifuged as previously described in the rEV isolation section. If indicated, an additional proteinase K (PK) (P6556-100MG, Sigma, Diegem, Belgium) treatment was performed after concentration and before density gradient centrifugation. This was done at a PK concentration of 1 mg/mL for 60 min at 37 °C and was ended at 4 °C. The concentration of 1 mg/mL was found optimal and reproducible after testing multiple PK concentrations on rEV spiked plasma (supplementary figure 13a, b). To isolate EV and rEV from the ODG fractions, a second SEC was included following previously mentioned protocol, unless stated otherwise. From this second SEC, eluted fractions 4-5-6-7 were pooled and analyzed with (f)NTA or ELISA.

**EV isolation from urine**. 50 mL of urine is centrifuged for 10 min at $1000 \times g$ and 4 °C after which the supernatant is concentrated ×100 with a Centricon plus-70 centrifugal filter with a nominal cut-off weight of 10 kDa (Millipore, Burlington, MA, USA) to obtain an 800 μL sample. This concentrated urine is then mixed with

3.2 mL OptiPrep to obtain a 40% iodixanol solution (4 mL). 20% (4 mL), 10% (4 mL) and 5% (3.5 mL) iodixanol solutions are added and finally 1 mL of PBS is layered on top of the gradient. This gradient is centrifuged for 18 h at $100,000 \times g$ and 4 °C and density fractions are collected and processed according to the rEV isolation section.

**Induced variability**. After isolation of EV from 2 mL of rEV-spiked plasma from the same pool (6×) via ODG centrifugation, resulting in a 4 mL sample containing rEV and EV, specific volumes of sample were replaced by PBS ($2 \times 1000$ μL and $2 \times 500$ μL) to simulate respectively 25% and 12.5% inter-user variability. Two samples were left unchanged. After this induced inter-user variability, total sample EV were quantified with NTA and the total rEV with fNTA.

**(Immune-) electron microscopy**. Isolated EV were deposited on glow-discharged formvar carbon-coated grids and stained with neutral uranylacetate and embedded in methylcellulose/uranyl acetate. For immune-electron microscopy, the grids containing the vesicles were incubated with 1% BSA in PBS blocking solution. Antibodies and gold conjugates were diluted in 1% BSA in PBS. The grids were exposed to the primary anti-CD63 antibody for 20 min, followed by secondary antibody rabbit anti-mouse IgG (Zymed, San Francisco, CA, USA) for 20 min and protein A-gold complex for 20 min. The grids were examined in a Tecnai Spirit transmission electron microscope (FEI, Eindhoven, The Netherlands). Images were captured by Quemesa charge-coupled device camera (Olympus Soft Imaging Solutions, Munster, Germany).

**Nanoparticle tracking analysis**. Nanoparticle tracking analysis (NTA) was performed using a NanoSight LM10-HS microscope (NanoSight, Amesbury, UK) equipped with a 45 mW 488 nm laser and an automatic syringe pump system. For conventional NTA three 30 s videos were recorded of each sample with a camera level of 13, a detection threshold of 3 and a syringe pump infusion speed of 20. For fluorescent NTA measurements (fNTA) an additional 500 nm longpass filter was used, and the camera level was increased to 16. Temperatures were monitored throughout the measurements, we assumed a medium viscosity of 0.929 cP and the videos were analyzed with NTA software 3.3. For optimal measurements, samples were diluted with PBS until particle concentration was within optimal concentration range of the NTA Software (3E8–1E9). For recovery calculations the number of fluorescent particles was measured before spiking. All size distributions determined with NTA correspond to the hydrodynamic diameters of the particles in suspension.

**High-resolution flow cytometry**. High-resolution flow cytometric analysis of rEV was performed on a jet-in-air-based BD Influx flow cytometer (BD Biosciences, Franklin Lakes, NJ, USA) using an optimized configuration[41]. In brief, a 200 mW 488 nm laser (Sapphire; Coherent, Santa Clara, CA, USA) and a large-bore nozzle (140 μm) were used, sheath pressure was permanently monitored and kept within 4.89 to 5.02 psi, and the sample pressure was set at 4.29 psi, to assure an identical diameter of the core in the jet stream. The BD Influx was triggered on the fluorescence signal derived from rEV, and thresholding was applied on this channel. All scatter and fluorescence parameters were set to a logarithmic scale. One hundred and 200 nm yellow green (505/515) and 100 nm red (580/605) Fluosphere beads were used to align the flow cytometer (Invitrogen, Carlsbad, CA, USA). To ensure that each measurement was comparable we loaded predefined gates and PMT settings. The threshold level was adjusted to allow an event rate ≤10/s when running clean PBS (supplementary fig. 14). Forward scattered light was measured with a collection angle of 15–25° (reduced wide-angle forward scatter [rw-FSC]). rEV stocks were diluted in PBS and vortexed just before measurement. EV counts were determined by measuring each sample within an experiment for a fixed amount of time (30–120 s). The event rate was below 10,000/s to avoid coincident particle detection and occurrence of swarm.

**Zeta potential measurements**. Zeta potential measurements were performed with a Zetasizer Nano ZS, which makes use of laser doppler electrophoresis, (Malvern Instruments Ltd, Malvern, UK) in disposable folded capillary cells at 22 °C in distilled water. EV and rEV were suspended in distilled water and five measurements of 10–100 runs were performed using the "automode" option. Zeta potential values given are the means of the five respective measurements.

**Refractive index determination**. NTA can be used to determine the RI of nanoparticles. To calibrate the scatter signals (supplementary figure 3b), 100 nm, 200 nm and 400 nm polystyrene beads (Malvern Instruments Ltd, Malvern, UK) were diluted in PBS. EV samples were thawed and diluted in PBS. We visualized scattered light from particles illuminated by a 65 mW 405 nm laser by a LM10 dark-field microscope (NanoSight, Amesbury, UK). For all samples, three videos of 30 s were captured with NTA software 3.3. Temperature was monitored during each measurement and we assumed a medium viscosity of 0.929 cP. Because the scattering power of the samples differs more than three orders of magnitude, each sample required different camera settings (supplementary table 3) to prevent pixel saturation. RI determination by NTA was accomplished by independently

measuring the diameter and light scattering power of individual particles with NTA and solving the inverse scattering problem with Mie theory. The data analysis in this manuscript was exactly done as described by Van der Pol et al.[8]. The scaling factor, which relates the theoretical scattering cross section to the measured scattering intensity, is 1.774 for this instrument.

**rEV quantification via fluorescence intensity measurement**. Fluorescence measurements were performed with a SpectraMax paradigm multi-mode microplate reader (Molecular Devices, Sunnyvale CA, USA) equipped with a 488 nm laser and a 500 nm filter. As a positive control an Alexafluor488 antibody (Life Technologies, Carlsbad, CA, USA) was used, and the relative fluorescence units (RFU) were calculated by subtracting the FU of a negative control (PBS) from the FU of the samples. All measurements were performed in triplicate.

**rEV quantification via anti-p24 ELISA**. Gag-EGFP protein concentrations were determined with the commercially available anti-p24 ELISA kit (Innotest HIV antigen mAB (Fujirebio, Ghent, Belgium). The assay was performed according to the manufacturer's instructions. For recovery calculations a rEV standard curve, from the same batch as used for spiking, was included ranging from 1E7 to 1E6 fluorescent particles as previously measured with fNTA.

**Protein analysis**. Protein concentrations of EV were measured, after lysis with 0.2% sodium dodecylsulphate (SDS) (L3771-500G, Sigma, Diegem, Belgium), with the Qubit Protein assay kit (ThermoFisher, Waltham MA, USA) and Qubit fluorometer 3.0 following manufacturer's instructions. Protein concentrations of cell lysates, obtained in Laemmli lysis buffer (0.125 M Tris–HCl [pH 6.8], 10% glycerol, 2.3% SDS), were determined using the Bio-Rad DC Protein Assay (Bio-Rad, Hercules, CA, USA). For protein analysis, samples were dissolved in reducing sample buffer (0.5 M Tris–HCl [pH 6.8], 40% glycerol, 9.2% SDS, 3% 2-mercaptoethanol, 0.005% bromophenol blue) and boiled at 95 °C for 5 min. Proteins were separated by SDS-PAGE and transferred to nitrocellulose membranes (Bio-Rad, Hercules, CA, USA). Membranes were blocked for 30 min in blocking buffer (5% non-fat milk in PBS with 0.5% Tween-20) and incubated overnight at 4 °C with primary antibodies. Secondary antibodies were added for 60 min at room temperature after extensive washing with blocking buffer. After final washing, chemiluminescence substrate (WesternBright Sirius, Advansta, Menlo Park, CA, USA) was added and imaging was performed using Proxima 2850 Imager (IsoGen Life Sciences, De Meern, The Netherlands). Quantification of signal intensity was performed using ImageJ software.

**LC-MS/MS analysis**. Volumes of rEV and mock EV samples in three biological replicates, each containing $5 \times 10^{10}$ particles in PBS as measured with NTA, were first reduced by vacuum drying to 1.5 mL. Amphipol A8-35 at 1 mg/mL was added to the samples, vortexed and incubated for 10 min at room temperature (pH 7). Next, the samples were reduced and alkylated with respectively 15 mM TCEP and 30 mM iodoacetamide for 15 min in the dark at 37 °C. The samples were acidified with 5% formic acid to pH 3 and centrifuged for 10 min at $16,000 \times g$. The resulting protein containing pellets were re-dissolved in 0.5 mL 50 mM ammonium bicarbonate pH 8.0 followed by overnight digestion with 2.5 μg trypsin at 37 °C. Following trypsin digestion, the samples were acidified to pH 3 resulting in Amphipol A8-35 precipitation. The samples were centrifuged for 10 min at $16,000 \times g$ at room temperature to pellet precipitated Amphipol A8-35, while peptides remain in solution. The supernatant containing the peptide material was concentrated by vacuum drying to 20 μl of which 8 μl was injected for LC-MS/MS analysis on an Ultimate 3000 RSLCnano system (Thermo Fischer scientific, Erembodegem, Belgium) in-line connected to a Q Exactive mass spectrometer (Thermo Fischer scientific, Erembodegem, Belgium). Trapping was performed at 10 μl/min for 4 min in trapping solvent (0.1% TFA in water/acetonitrile (98:2, v/v)) on a 100 μm internal diameter (I.D.) × 20 mm trapping column (5 μm beads, C18 Reprosil-HD, Dr. Maisch, Germany) and the sample was loaded on a reverse-phase column (made in-house, 75 μm I.D. x 150 mm, 3 μm beads C18 Reprosil-HD, Dr. Maisch). Peptides were eluted by a linear increase from 2 to 55% solvent B (0.1% formic acid in water/acetonitrile (2:8, v/v)) over 120 min at a constant flow rate of 300 nl/min. The mass spectrometer was operated in data-dependent mode, automatically switching between MS and MS/MS acquisition for the 10 most abundant ion peaks per MS spectrum. Full-scan MS spectra ($400–2000$ $m/z$) were acquired at a resolution of 70,000 in the orbitrap analyzer after accumulation to a target value of 3,000,000. The 10 most intense ions above a threshold value of 17,000 were isolated (window of 2.0 Th) for fragmentation at a normalized collision energy of 25% after filling the trap at a target value of 50,000 for maximum 60 ms. MS/MS spectra ($200–2000$ $m/z$) were acquired at a resolution of 17,500 in the orbitrap analyzer. The S-lens RF level was set at 50 and we excluded precursor ions with single, unassigned and charge states above 5 from fragmentation selection.

Data analysis was performed with MaxQuant (version 1.5.4.1) using the Andromeda search engine with default search settings including a false discovery rate set at 1% on both the peptide and protein level. Spectra were searched against the human protein entries in the Swiss-Prot database (downloaded from http://www.uniprot.org/, version from May 2016 containing 20,195 human protein sequences) supplemented with the sequence of the gag-EGFP fusion protein. The

mass tolerance for precursor and fragment ions were set to 4.5 and 20 ppm, respectively, during the main search. Enzyme specificity was set as C-terminal to arginine and lysine, also allowing cleavage at proline bonds with a maximum of two missed cleavages. Carbamidomethylation of cysteine residues was set as fixed modification. Variable modifications were set to oxidation of methionine residues and acetylation of protein N-termini. Only proteins with at least one unique or razor peptide were retained leading to the identification of 2324 proteins. Proteins were quantified by the MaxLFQ algorithm integrated in the MaxQuant software. A minimum ratio count of two unique or razor peptides was required for quantification. Further data analysis was performed with the Perseus software (version 1.5.4.1) after loading the protein groups file from MaxQuant. Reverse database hits and contaminant proteins were removed and LFQ protein intensities were log2 transformed. Replicate samples of both conditions were grouped and proteins with less than three valid values in at least one group were removed and missing values were imputed from a normal distribution around the detection limit. Then, a $t$-test was performed for pairwise comparison of both conditions. The results of this $t$-test is shown by the volcano plot in Supplementary Fig. 4a. For each protein, the log2 (rEV/mock EV) fold change value is indicated on the $X$-axis, while the statistical significance ($-$log p-value) is indicated on the $Y$-axis. All raw data were submitted to the PRIDE database (PXD010269).

**Immune precipitation.** MagnaBind goat anti-mouse IgG or goat anti-rabbit IgG magnetic beads (Thermo Fischer scientific, Erembodegem, Belgium) were incubated with 10 µg anti-CD81, anti-CD63, anti-PEG antibody or PBS (negative control) per 200 µL beads for 2 h at 4 °C while rotating. Beads were washed three times with 500 µL PBS supplemented with 0.001% Tween20 (Sigma, Diegem, Belgium) and incubated with the sample containing rEV or rEV-PEG for 2 h at 4 °C while rotating. Beads were washed three times and the supernatant was pooled and pelleted using an SW55 rotor (Beckman Coulter, Brea, CA, USA) at 100,000 × g for 70 min. Pellets were lysed with reducing sample buffer.

**RT-qPCR gene expression analysis.** Total RNA was isolated from vesicles using the miRNeasy Micro kit according to manufacturer's instructions (Qiagen, Valencia, CA, USA), and eluted in 14 µL DNase/RNase-free water. Ten microlitre of RNA was reverse transcribed in a 20 µL reaction using iScript cDNA Synthesis Kit (Biorad, Hercules, CA, USA) according to the manufacturer. EGFP mRNA expression analysis in rEV was performed via reverse transcription quantitative polymerase chain reaction (RT-qPCR) using the LightCycler 480 SYBR Green I Master kit (Roche Applied Science, Penzberg, Germany). The 5 µL PCR reaction mix contained reverse and forward primers (0.5 µL of a 5 mM stock solution), LightCycler 480 SYBR Green I Master (2×) (2.5 µL) and cDNA (2 µL of a ¼ dilution corresponding to the cDNA reverse transcribed from 10 µL RNA). The 384-well plate was then run on the LC480 (Roche Applied Science, Penzberg, Germany) at 95 °C for 5 s, then 60 °C for 30 s and 72 °C for 1 s (for 44 cycles). Following primers were used: EGFP primer pair 1 with forward sequence GAC-GACGGCAACTACAAGAC and reverse sequence TCCTTGAAGTC-GATGCCCTT and EGFP primer pair 2 with forward sequence TAAACGGCCACAAGTTCAGC and reverse sequence GAACTTCAGGGTCAGCTTGC.

**Lipid analysis.** Lipids were extracted from 5 x 10¹⁰ rEV and mock EV in three biological replicates, as measured with fNTA, using a modified Bligh-Dyer protocol and phospholipids were analyzed by electrospray ionization tandem mass spectrometry (ESI-MS/MS) on a hybrid triple quadrupole linear ion trap mass spectrometer (4000 QTRAP, AB SCIEX, Framingham, MA, USA) equipped with a robotic sample injection and ionization device (TriVersa NanoMate, Advion, Amsterdam, The Netherlands). The collision energy was varied as follows: prec 184, 50 eV; nl 141, 35 eV; nl 87, −40 eV; prec 241, −55 eV. The system was operated in the multiple reaction monitoring (MRM) mode for quantification of individual species. Total cholesterol concentrations were measured with the Amplex Red Cholesterol Assay Kit (A12216, Thermo Fischer Scientific, Erembodegem, Belgium) following manufacturer's instructions.

**rEV PEGylation.** rEV (2 x 10¹⁰) were incubated in 100 µL of 0.016 mg/mL DMPE-PEG 5k (PG1-DM-5K, Nanocs, NY, USA) at 40 °C for 2 h while being gently mixed. This concentration of DMPE-PEG 5 K was optimized using a DMPE-PEG-biotin construct (Nanocs, NY, USA) and precipitation using streptavidin coated magnetic beads. Using western blot analysis targeting EGFP we saw the highest signal in the precipitation and the lowest signal in the flow through at a concentration of 0.016 mg/mL DMPE-PEG-biotin (supplementary fig. 13c).

**Lyophilization.** For lyophilization, samples were diluted 1:20 in 100 µL PBS containing 5% trehalose to a final concentration higher than 5 x 10¹⁰ particles/mL. The vials were placed on pre-cooled shelves at -45 °C for 2 h after which the chamber pressure was lowered to 0.1 mbar. From the moment the desired pressure was reached, the shelf temperature was increased at 1 °C/min to −25 °C and maintained for 24 h. When all ice was completely sublimated the temperature was increased at 0.15 °C/min to the final drying temperature of 20 °C and this temperature was maintained for 4 h. Finally, the temperature was decreased again at 1 °C/min to a storage temperature of 3 °C while maintaining the vacuum until the cycle was stopped.

**Statistical analysis.** All statistical analyses performed in this manuscript were done using GraphPad Prism v7.

**EV-TRACK.** We have submitted all relevant data of our experiments to the EV-TRACK knowledgebase (EV-TRACK ID: EV190040)[4].

**Reporting summary.** Further information on research design is available in the Nature Research Reporting Summary linked to this article.

## Data availability

Proteomic data were submitted to PRIDE database (PXD010269). All relevant data of our experiments were submitted to the EV-TRACK knowledgebase (EV-TRACK ID: EV190040). The source data underlying Figs. 2, 3, 4, 5 and supplementary figures 1, 2, 5, 6, 7, 8, 10, 11 and 13 are provided as a source data file. All other relevant data that support the findings of this study are available from the corresponding author upon reasonable request.

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

## Acknowledgements

This work was supported by Ghent University (Concerted Research Actions and Industrial Research Fund), Ghent University Hospital, Kom Op Tegen Kanker, post-doctoral position and Krediet aan Navorsers (A.H.) and PhD position strategic basic research (E.G.) from Fund for Scientific Research Flanders (FWO), by the European Union's Horizon 2020 research and innovation programme under the Marie Skłodowska-Curie grant agreement No 722148 and by the European Research Council under the European Union's Seventh Framework Programme [FP/2007–2013]/ERC Grant Agreement number [337581]. E.V.P.D. was funded by the Netherlands Organisation for Scientific Research—Domain Applied and Engineering Sciences (NWO-TTW), research program VENI 15924.

## Author contributions

E.G. planned, designed and performed the experiments and co-wrote the manuscript. J. T., B.D., J.V.D., L.L. and G.V. performed EV separations and characterizations from various cell cultures and biofluids. E.H. contributed to the optimization of rEV PEGylation. D.D.S. maintained bacterial cultures for plasmid purification. K.G. and F.I. performed and analyzed proteomics experiments. I.M. performed electron microscopy. P.J. V.B. and T.D.B. performed lyophilization. M.W. and E.N.H. Performed and analyzed HR-FC experiments. K.B. and J.S. performed and analyzed lipidomics experiments. E.V. D.P. and R.N. performed RI calculation. G.B. and H.D. collected patient material. N.C., P.M., J.V., H.D. and S.E. assisted in experimental design. O.D.W. and A.H. initiated the project, designed and interpreted experiments and co-wrote the manuscript. P.M., J.V., S. E., O.D.W. and A.H. conceived the rEV technology. All authors read and approved the manuscript.

## Additional information

**Competing interests:** A.H., O.D.W., S.E., J.V., P.M. and E.G. are inventors on the patent application covering the rEV technology (WO2019091964). The remaining authors declare no competing interests.

