## [Peer Review File · Nature Communications]

Reviewers' comments:

Reviewer #1 (Remarks to the Author):

This well-written manuscript describes well-designed and executed experiments to characterize a proposed, much-needed extracellular vesicle standard. Such a standard would be useful for the community. Especially important is the PEGylation approach, facilitating spike-in studies. The materials and methods section is crucial to the manuscript, and as it commendably appears to contain the details needed for reproduction and replication studies, this reviewer does not have extensive comments. Statistical analyses appear to be appropriate. One set of experiments comes to mind, though, that in my opinion would strengthen the manuscript.

Major point

The finding that EVs (or rEVs) from HIV-1 gag-overexpressing cells have the same density as EVs from control cells or biological fluids is somewhat surprising to me. The finding may be due to the use of a wider range of densities in some separation gradients than are generally used in the HIV field, although the flotation gradient experiments might have taken care of this possibility.

Alternatively, it could be that there is relatively little gag incorporation compared with previous estimates of gag in HIV-1 VLPs/virions, consistent with the EMs that do not seem to show a strongly electron dense shell. I would recommend considering at least two of the following:

1. Perform more narrow density gradients, like those used in the HIV literature to separate HIV from EVs, starting with Dettenhofer and Yu, 1999 J Virol.
2. As a control to be sure the techniques can efficiently separating VLPs from lighter EVs, I would recommend comparing HIV particles with the rEV VLPs: are there differences? Can they be separated? The HIV particles don't have to be infectious, just more complete than rGag VLPs alone.
3. Check the stoichiometry of gag per particle considering the EM results. Maybe there is less gag in these rEVs than in previous virus or VLP estimates as cited in the ms?

Minor points

A. Figures: dot plots or box and whiskers etc. would be preferred to bar graphs throughout.

B. I am not sure that the term "recombinant EV" is ideal, although it is catchy and neat. By the definition used here, any EV made from a transfected or otherwise altered cell would be "recombinant." I view these EVs rather as VLPs that contain a recombinant protein. The authors are urged to reconsider the term, although perhaps they will conclude that they simply disagree with me.

C. Please also add the reference to the Eyckerman article in the materials and methods section (pMET7 plasmids) for ease of reference for the reader.

D. Use "nanoparticle" instead of "nano particle"

-Kenneth W. Witwer

In the interest of transparency, I am a co-author on several position papers of the International Society for Extracellular Vesicles as well as the EV-TRACK article (van Deun, et al, Nat Methods, 2017) with several of the authors of the manuscript under review. However, I do not consider these to be conflicts of interest, since these articles have involved large numbers of authors, I do not have any active research collaboration with the authors, and I have not been involved in this project.

Reviewer #2 (Remarks to the Author):

The manuscript of Geurickx et al (NCOMMS-18-36122-T) tackles a burning issue in the research field of extracellular vesicles (EVs): how to overcome the disparity in the currently used methodology and henceforth the generated results, which sometimes vastly differ even when produced with the same analytical method. For this purpose, the authors developed a fluorescent

reference EV material by transfecting a plasmid coding for Enhanced Green Fluorescent Protein coupled to the HIV-1 gag protein into HEK293T cells which resulted in the formation of virion-like particles, which due to the hijacking of the ESCRT-mediated route of exocytosis, budded from similar membrane foci as native EVs. The resulting rEVs were fluorescent and could be tracked more easily than native cell-derived EVs and also due to the external cargo (gag/EGFP), they could be specifically detected or isolated from the whole EV pool by e.g. western blotting and immunocapture, respectively.

Moreover, to reduce the build-up of a protein corona, which was observed when plasma (but not urine or cell medium) was spiked with rEVs, they showed that the rEVs could be PEGylated to diminish this effect, increasing the field of applicability for rEVs as a potential standard. rEVs were spiked e.g. into biofluids to compare isolation efficacies of different methods and CV% of methods to evaluate intra-assay repeatability or inter-operator effects. The authors have produced a huge amount of experimental data to show the applicability of the rEV and also to provide information on rEV stability and reproducibility of its use.

The aim of the study is highly commendable. A useful reference material, and especially a SOP enabling open production of such a material in all laboratories, as promised in the abstract, would be welcomed by the EV research community. Availability of a reliable reference material could be a game changer in the development of the EV field enabling reliable comparison of results between laboratories. Increasing transferability of data between laboratories and reproducibility of methods is a key prerequisite for the development of EV use in diagnostics and therapeutics. The study seems to be carefully executed and the statistical data sound, as far as my expertise covers the used methods. There are no ethical concerns.

However, in its present format the data package is extremely large and complex and presented in a manner, which makes it hard to read. There are inconsistencies in the manner of reporting, which make it difficult to decipher the results or judge the validity of the claims. Therefore, to be considered for publication, the manuscript will require an extensive and systematic revision involving control experiments, rewrite-up and also consideration of how the data is best displayed better. I also have some concerns on the data interpretation, which may be due to the scarcity of the text and the unsystematic manner of reporting. However, to do the data justice and improve readability of the paper, these issues need careful addressing. In some cases, I have asked for further experimentation, and these experiments are not difficult and will hopefully increase the reliability of the critical results, as it is possible that this could become a milestone paper. I hope that the authors will find my lengthy and thorough review helpful in improving the paper to enable its acceptance for publication.

Technical comments (major):

1. There are just two main figures in the manuscript and they are (too) full of data generated by different methods. Also, the number of supplementary figures is big, and when the same data are presented in several figures or the data on the same topic is discussed in a different manner in different places (e.g. buoyant density in several figures). As the main results text is very scanty, the results need to be read jointly with the figures. It would be easier for the readers to follow and interpret the data if they were depicted in a few more main figures grouped around the main findings. In my opinion, the main findings are the production (SOP: what and how), characterization (results of rEV properties mentioned in bulk text and then explored and grouped into the supplementary material in detail) and the various examples of use (each with their own dataset with either figure or table). However, this is up to the authors, and they may find a better way to restructure their data.

As a suggestion to reduce the supplements and recombine some of the current data e.g. Supplementary Fig 11 could be combined with the main Figure 2 panels a-d and Supplementary figure 12 with the main figure 2 panels e-j.

2. Spiking:

There needs to be an explanation and some discussion of how and why the rEV use would allow a

significant difference to be detected when it was not measured without the rEV use. It is well known that the plasma of cancer patients has abnormal plasma proteins e.g. acute phase proteins. Could e.g. a corona effect due to the plasma of the cancer patients change the isolation of the rEV and thereby falsely higher values of EVs? (see comment 3 below)

Was the PEG-rEV used in this study or the unpegylated form? If the unpegylated, please verify the result with the PEG-rEV (see comment 3 below).

"As an example, sample EV and spiked rEV from plasma of breast cancer patients (n=26) and healthy volunteers (n=11) were isolated by SEC-ODG-SEC, and were measured by NTA and fNTA. The average, normalized sample EV concentration per mL plasma was increased by 2.5 fold (5.83×10^{10} vs 1.46×10^{11}) ($p=0.0002$, Mann-Whitney test) in breast cancer patients compared to healthy individuals." Please use past tense. Please, switch the order of the numbers as now the breast cancer EV amount seems lower than the healthy controls. Was this result measured by NTA or fNTA? Why only one set of results shown, if both methods were used?

3. Changes in the buoyant density of the rEV when in plasma:

Since it is crucial for the use of rEV reference as a spike in, the results of the enhanced detection are important, but rather curious, and thus, I would suggest the following controls to be done: Using WB or flow to identify

- density position in ODG of plasma-incubated rEVs after PK-treatment (first plasma, then PK)
- position in OGD of native plasma EVs after PK-treatment
- position in OGD of rEVs treated with PK before adding to plasma

compared by samples shown in SFig10 panel b. Also, whether the source of plasma from normal donors or cancer patients affects the detection could be tested by analyzing the shift of rEV (or PEG-rEV if it was used in the original exp) in the ODG after incubation in the relevant plasma pools from normal donors and cancer patients.

4. Patient plasma collection:

Age and sex are known to effect plasma EV concentrations: were the controls matched for age and sex? Of course, 26 is no match for 11, but since age and sex influence, were there males in the controls and was the age similar. If not matched at all, state this in the text as "unmatched controls (sex, age)".

Were the healthy controls and the patients treated similarly regarding fasting prior to the blood draw and the time of blood draw?

Was the plasma verified to be platelet-free after double-centrifugation at 15 min 2500 x g? If not, omit the unnecessary and likely false claim of "platelet-free".

5. Induced variability:

Abbreviated titles in the table do not explain well enough what was done experimentally and how the calculations based on these results were made. Please, provide more detail of the measurements, procedure and calculations in the main text, or in the M&M section, or in the legend.

If 1000 or 500 ul of 2000ul plasma sample were replaced with PBS, it is 50% not 25%, and 25% not 12.5%, respectively?

6. Were equal number of replicates analysed with each of the methods when the CVs were calculated? Please state the exact number of the replicates for each methods in the text.

E.g. Using rEV, the intra-method CV for the EV isolation methods SEC, SEC-ODG-SEC and dUC are respectively 5.7% (n= x), 9.2% (n= y) and 18.0% (n= z) as calculated by fNTA. The intra-method CV for rEV quantification methods fNTA, RTqPCR and anti-p24 ELISA are respectively 5.8% (n= x), 0.8% (n= y), and 5.7% (n= z), (figure 1i,j). When CVs are compared they should be derived from

the same amount of measurements for each method.
<https://www.ncbi.nlm.nih.gov/pmc/articles/PMC4478151/>

7. How much fluorescence from rEVs was present in the other density fractions (ie in other vesicle types) than those selectively collected for the standard?

8. If the HEK293T cells (only?) were used for rEV production, it should be stated in the methods. Was the transfection stable or done anew for each isolation? Data in Supplementary Fig 2 could be combined into Supplementary Fig 1, panel b. In my mind, only the repeatability of the isolation (panel c) is relevant information and the rest of the results could be just mentioned in the text. Please, state in the text and M&M from which of the tested cell lines the rEVs for the studies were isolated from. Also, the source of EVs E.g. Figure 1 panel I is missing.

9. What does this mean: "plasma was diluted with equal volumes(?) of PBS..." Please, clarify in the text if this means 1:1 plasma:PBS. Was equal volume of (PFP)-plasma taken from all study subjects? How much plasma (vol) was then processed for each EV patient/control sample?

10. In Figure 1, the number of biological and/or technical replicates is missing in b, c, d, e, f, k, m, n, l. Please add this info similarly as it is stated in h-j, unless it is not applicable. Also, please add the relevant method for b (NTA or fNTA). Why is the size distribution of rEV different in Fig 1b from 2b (text 105 nm +/- 9)?

Panel g. Please, show the position of the Fluorosphere bead-based gates and noise. Although not explained, I think this info is partially displayed in Supplementary Fig 9.

Panel n: add PK-treated plasma in the figure.

Main Fig 1 electron microscopy images could be larger or with a better resolution: from k it is impossible to see any details. As there is a Supplementary figure 4 for EM images showing practically the same dataset, EM images could be removed from Figure 1 to reduce the data heavy figure and panel k could also be shown larger here. Please rephrase the wording "share morphology" as the morphologies seem to be different in all the EV populations shown. It would also be beneficial for comparison to have the different EVs shown at the same magnification (esp. 4T1). Please show how do the lyophilized and reconstituted rEVs perform in ODG compared to "fresh" rEV? Or could the authors show another, more reliable way for proving intactness of the rEVs post lyophilization? (see comment 17)

11. All information of the plasmid pMET7-gag-EGFP is missing from the M&M. By Googling, I found that the plasmid has previously been published by one of the co-authors and that there is a citation recommended for its use. Please provide the relevant information for this in the text. "pMET7-GAG-EGFP was ... (Addgene plasmid # 80605 ; <http://n2t.net/addgene:80605> ; RRID:Addgene_80605)". Further, the schematic of the plasmid structure could be placed in Figure 1 a, as it is currently written as to contain this info.

12. Supplementary Figure 3:

Why is the data for rEV behavior in ODG given in three separate figures? ODG is indeed an important way to compare the similarity of rEV and native EVs. However, Figure 3 is confusing, because all the different EVs are reported in a dissimilar manner regarding the densities making it very hard to compare. Also, different EV markers are used for the EVs. Please, provide a consistent data set for the same density bins and using the same set of markers. I am particularly concerned about the plasma data. Most likely the EV number/isolation was too low?

13. Supplementary Figure 5: Could the authors comment on the proteins that were found to be unique for rEV or the mock-transfected EVs. Any pathway differences?

14. Supplementary Fig 6. As the lipid class percentages for rEV were stated in the text, it would be interesting to add in the same parenthesis also the percentages found for the mock- transfected

EVs for comparison.

15. Supplementary Figure 7: Two data sets by different methods are presented, but what is the significance of comparing the different RI of different EVs to this extent? How can the RI information be used? The RI of the most interesting species regarding rEV use as a spike-in (i.e. the plasma EVs) is missing. Please, add. Please rephrase the title ("share refractive index"). As there clearly seems to be challenges using flow cytometry for the RI determination (due to the lower limit of detection? Also, the x axis of panel B does not show well, please state values for comparison in the fig., if possible), could the dataset in Fig 7 be reduced by omission of the flow data? Not being an expert, I would like to know does the fluorescence from EGFP in rEV (up to 5000 copies in VLP according to introduction) influence the determined RI? Or is EGFP not excited at the relevant wavelengths in the used methods? Please, comment. The number of biological and technical repeats is missing.

What is the relevance for Supplementary Figure 14? Could this be replaced with a reference or combined with Supplementary Fig 7?

16. Supplementary Figure 8 panel D. Please increase the font size for the rEV concentrations in the figure. How was the concentration measured? fNTA or NTA?

17. Supplementary Fig 9:

Please also provide the one year storage data, or explain better in the legend when FT1 and 2 measurements were done. Could panel D be provided as overlapping plots to ease comparison? Alternatively, add the mean and the mode information in the figure. Panel e: Please provide the event counts before and after lyophilization in the figures. It seems that there are more counts after lyophilization in the detectable range, which could indicate break-up/fusion? Please, discuss. See also comment 10.

18. P-values: I would find reporting the nonsignificant values in the case of negative results (e.g. Fig 9a-c storage data, SFig 2b viability data etc) useful.

Technical comments (minor):

1) How was the freezing of samples/rEV performed? (snap or?). How was thawing performed? Please, add this information.

2) "Efficiencies of SEC, two-dimensional size- and density-based separation (SEC-ODG-SEC)²⁰, and differential ultracentrifugation (dUC) to isolate sample EV from plasma were respectively nearly 100%, 30% and 10% (figure 2 e, f)." Should this be rEV? Also, once something is abbreviated (SEC-ODG-SEC, please use the abbreviation also later in the text.

3) For the following assays data concerning the sample numbers/sample size/used material (volume/particle number/ protein) for each assay would be important information:

Protein mass spectrometry

Lipid mass spectrometry

Total cholesterol

ELISA

Related comment: Please elaborate in the text what this means: "To quantify the total amount of phospholipids, we summed the abundances of individually measured species within each phospholipid class and normalized based on the amount of protein."

4) Immuno-EM: Please add info of the used dilution of primary antibody (anti-CD63) and the secondary. Gold size, dilution and the commercial source for protein A are also missing. Fig 1 legend is misleading, because it describes the use of a gold-conjugated antibody, which is not the case according to M&M?

5) Supplementary table 2: please add the word EV where appropriate.

Content & writing (major):

1. The abstract states that a SOP will be communicated in the ms. However, after reading the main text & figures, it is unclear for me what is the SOP and what is it provided for? Please clarify

in the text (aim & discussion), and also e.g. by mentioning in Supplementary Figure 1, if that describes the intended SOP for rEV production? Regarding the SOP, it should also be clearly defined in the M&M as now the production of rEV and isolation of EVs from different sources run together.

2. The manuscript is clearly and fluently written, but it has a large amount of grammatical errors regarding e.g. missing punctuation (participial phrases such as "To understand and mitigate these limitations, we have..."). Most importantly, the use of the present tense when describing results of own data leads to difficulties in separating pre-existing information, accepted knowledge and facts from the results generated in the experiments of this study. I strongly urge changing the verbs in the description of any experimentation into past tense. E.g. in the abstract "Implementation of rEV reduce**d** intra-method and inter-user variability of EV sample preparation and analysis, and improve**d** the sensitivity of EV enumeration in biofluids." states an observation, whereas in the present tense, as it is now in the text, it sounds like a statement or fact.

3. The use of "commutable" in the abstract and at several places in the text as an adjective is not easily understood for the indicated meaning of the rEVs' analogous behavior with native EVs under various experimental conditions. Please rephrase or -word.

4. The use of word "share" with morphology/ size /density etc. "Had similar" or "did not differ" would be more appropriate.

Minor comments:

What does the informed use of rEV mean? Informed by whom? Could the authors elaborate their meaning or remove.

What does "accurately measured" mean? Wouldn't accuracy require a standard, which is currently being developed in the study? Please rephrase or remove.

behaved in a linear manner?:

"rEV **behave linear** in defined concentration ranges when diluted in PBS and measured by a fluorescence microplate reader, densitometry of EGFP western blot bands and an ELISA for p24, a subunit of the gag polyprotein"

was/could?:

The latter **being** the most sensitive as it **can** detect rEV in a range of 10⁶-10⁷ rEV

What is size fraction?:

rEV eluted in the same **size fractions** as sample EV upon isolation from cell culture medium, urine and plasma using size exclusion chromatography (SEC)

What does inducible mean here? : "This shift was reversible and inducible."

"Treatment **of what?** with proteinase K (PK) before gradient separation reverses the density shift..." Plasma or rEV?

Please, deabbreviate EGFP at the first mention. Please, also check the abbreviated material throughout the manuscript, i.e. do not abbreviate if you do not use the abbreviation.

5. References:

- "Current reference materials for calibration such as polystyrene beads, silica beads or liposomes lack EV-like properties, resulting in an aberrant refractive index (RI) and inaccurate measurements^{7,8}." Do these references cover liposomes? If not, remove.
- "However, EV research and its biomedical applications are hampered by the myriad of isolation

and measurement methods and the lack of appropriate reference materials for accurate calibration, normalization and method development⁴⁻⁶.”

Please, change ref 5 to the newest MISEV guidelines Thery & Witver JEV2018, where the most recent understanding of the complexity of the EV field regarding isolation, lack of reference materials, etc is addressed.

- “The suitability of HIV-1 VLP as a reference material for EV research has not been previously considered or tested.” This is true and important novelty of the study. However, although I am not an expert, retroviral protein-aided transfer of protein has been previously described as a strategy for other purposes with EVs e.g. vaccination platforms in the literature/patents. Discussion and referencing of this in regard to the development in the field would be nice. I leave the choice of the examples to the discretion of the authors.

- Patent search by the indicated patent number did not produce any results. Please, verify the number.

Reviewer #3 (Remarks to the Author):

The authors suggested that they successfully generated recombinant EV as a trackable biological reference material for EV experiments. As the authors claimed, the use of reference EV is important in the field of extracellular vesicle research. Thus, the attempt and aim of the manuscript itself is noteworthy. However, there are points that the authors should explain or prove to claim that the rEV generated in this research can be a reference material for sample EV.

1. The rEV was produced from transiently transfected HEK293T cells. Thus, it is hard to believe that all the cells produce rEVs. This indicates that there should be the heterogeneous mixture of EVs (rEVs and EVs) in the culture medium. Even in the transfected cells, rEV production abilities among cells are different that native EVs as well as rEVs can be also produced at the same time. Some of the analytical methods used in this research cannot discriminate rEVs from EVs, there is a lack of evidence for most of the analytical data comparing rEVs with sample EVs, which significantly weakens authors' conclusion.

2. Although there are results showing rEV can be used as a reference to EV, there is no direct evidence showing why rEV is superior to other reference materials.

3. In addition, there are lots of research articles about the use of exogenously or endogenously labeled EVs with fluorescent molecules. What will be the potential issues for using them as reference materials? In this context, what is the novelty of the rEV?

4. There is no information on the amount of spiked rEV into each solution. In supplementary table 1, it is only mentioned that 1E10 rEV is spiked into plasma (to what volume?).

The amount of spiked rEVs and the ratio between spiked rEVs and already present EVs are important especially in the downstream analysis of EV biomolecules including proteins, RNAs, DNAs, etc. This is because of rEVs are also produced from the cells that they contain most of the biomolecules that sample EVs have. Thus, if there are too many rEVs spiked into the samples as compared to already present EVs in the sample, some of the specific biomolecules from sample EVs can be over- or under-estimated. This will critically limit the use of rEV as a reference material in the biomedical fields. In addition, the concentrations of sample EVs are entirely different among the bodily fluids including plasma and urine. So, for the practical use of rEV, the effect of amount and ratio of rEVs should be carefully studied to ensure its use as a reference material in this field. And as minor comments, there are duplicates of supplementary table 1 title. And the CV value after normalization in this table is 5.94% while it is described to be 6.1% in the text.

Based on these, although the experiments were well designed and the results themselves were well presented, the article does not yet satisfy the standard of Nature Communications.

Responses to the referees' comments

Reviewer 1

Major comments:

1. Perform more narrow density gradients, like those used in the HIV literature to separate HIV from EVs, starting with Dettenhofer and Yu, 1999 J Virol.

2. As a control to be sure the techniques can efficiently separate VLPs from lighter EVs, I would recommend comparing HIV particles with the rEV VLPs: are there differences? Can they be separated?

The method by Dettenhofer and Yu was adapted to the conditions available in the lab. Taking into account the k-factor, the SW31.1 rotor has a max of 187,600 g for 1 h 56 min corresponding to the conditions in Dettenhofer and Yu (250,000 g and 1 h 30 min)¹. We performed this Optiprep velocity gradient (OVG) centrifugation on both the concentrated conditioned medium (CCM) of the gag-EGFP transfected cells and on non-infective HIV-1 particles. One mL OVG fractions starting from the top to the bottom were evaluated by western blot analysis for EV-enriched protein syntenin-1, gag-EGFP and p24 (see figure below and **supplementary figure 6 a** in the revised manuscript). Syntenin-1 positive/gag-EGFP negative EV were detected in OVG fractions 4, 5 and 6 and thus separated from syntenin-1 positive/gag-EGFP positive rEV identified in OVG fractions 8 to 15. In agreement, rEV sedimented to equivalent OVG fractions as p24 positive non-infective HIV-1 particles (see figure below).

We further characterized OVG fractions enriched for rEV (OVG fractions 10 to 13) and OVG fractions enriched for endogenous EV (OVG fractions 4-5 and 6-7) using fNTA, western blot, mass spectrometry-based proteomics and electron microscopy (**Supplementary figure 6 b-e, j** of revised manuscript). Quantification by fNTA of rEV retrieved from OVF fraction 10-13 identified nearly 100 % fluorescent particles vs 79 % in rEV preparations obtained by equilibrium ODG centrifugation (**Supplementary figure 6 c and d**). This, together with the identification of a syntenin-1 positive/gag-EGFP negative EV in OVG fractions 4-5 and 6-7 confirms that rEV obtained by equilibrium ODG centrifugation contain approximately 20 % of contamination with endogenous EV produced by HEK293T cells (**supplementary figure 6a**). Furthermore, rEV separated by OVG centrifugation from medium conditioned by HEK293T cells transiently transfected with gag-EGP, showed the characteristic cup-like morphology (electron microscopy), contained all frequently studied EV-associated proteins (shown by

western blot and mass spectrometry based proteomics) and floated to the density characteristic for EV using the equilibrium ODG centrifugation.

We also analysed whether rEV from OVG fractions 10-13 could still be implemented for the applications presented in the manuscript. rEV prepared by OVG centrifugation were still detected in a linear manner using a fluorescent plate reader and the p24 ELISA and correlated semi-logarithmically with Cq values for EGFP mRNA targeted RT-qPCR (**supplementary figure 6f, g and h**).

In conclusion, OVG centrifugation enables the separation of rEV from endogenous EV released by gag-EGFP transfected HEK293T cells and results in a nearly 100% pure population of rEV. We thank the reviewer for this suggestion since it significantly improved the specificity of the product.

3. Check the stoichiometry of gag per particle considering the EM results. Maybe there is less gag in these rEVs than in previous virus or VLP estimates as cited in the ms?

We tried to address this concern by performing fluorescence correlation spectroscopy (FCS). With this method, the number of fluorescent particles within the confocal volume can be determined. By comparison of the number of particles before and after disruption of the particles, it would be possible to calculate the total amount of gag-EGFP molecules present in one rEV particle. However, within the current time frame we didn't succeed in the complete dissociation of the gag-EGFP molecules. Under all tested experimental conditions, a fair amount of complexed gag-EGFP aggregates was still detected, making determination of the number of individual gag-EGFP molecules impossible and thereby the calculation of the total amount of gag-EGFP molecules present in one rEV particle. The strong interactions taking place between the CA domains of the different gag proteins are probably the reason for this phenomenon.

This technology needs substantially more optimization which was not feasible within the time frame of the revision. We decided not to include any of these results in the manuscript. In agreement with the suggestion of the reviewer, we omitted the claim of 2500-5000 gag molecules per particle in the introduction.

Concerning the EM results and why no electron dense core is noticed; we would like to notify that this should not be visible in virus like particles (VLP) as in these particles the gag polyprotein is not cleaved, hence is staying at the membrane in contrast with mature HIV particles, where the electron dense core is made by cleavage of the gag sub-domains, more particularly, the CA domain. One would expect an electron dense shell at the membrane, but this we do not observe. Literature data support this observation since other research groups have similar findings². However, we do expect that by using cryo-EM this shell should become visible, as was also noticed by other research groups³.

Minor comments:

A. Figures: dot plots or box and whiskers etc. would be preferred to bar graphs throughout.

We believe that the bar charts used throughout the manuscript give a better representation than box plots because the density distributions, for what the bar charts are used for, have a gaussian-like distribution. In other experiments we explicitly used box/dot-plots. In agreement with the reviewer suggestions, we did change the bar charts for the recovery determination in box-plots (**figure 5 a, b, c** of the revised manuscript).

B. I am not sure that the term "recombinant EV" is ideal, although it is catchy and neat. By the definition used here, any EV made from a transfected or otherwise altered cell would be "recombinant." I view these EVs rather as VLPs that contain a recombinant protein. The authors are urged to reconsider the term, although perhaps they will conclude that they simply disagree with me.

We would rather keep the term recombinant EV (rEV) because these particles can, according to the MISEV2018 guidelines, definitely be called EV as this is a "generic term for particles naturally released from the cell that are delimited by a lipid bilayer and cannot replicate". The addition of "recombinant" makes sure that these EV are not arising in normal conditions, but that some engineering had to take place to create these fluorescent vesicles. As it is in the end also the goal to distribute rEV among different research groups across the world it is necessary to have a catchy name that people will remember and easily talk about with other researchers. This could hopefully spread the use of rEV and hereby also increase the stringency of obtained results throughout the EV field.

C. Please also add the reference to the Eyckerman article in the materials and methods section (pMET7 plasmids) for ease of reference for the reader.

The reference is added to the materials and methods section (p20 in the revised manuscript)

D. Use "nanoparticle" instead of "nano particle"

This has been adapted throughout the manuscript.

Reviewer 2

Technical comments (major)

1. There are just two main figures in the manuscript and they are (too) full of data generated by different methods. Also, the number of supplementary figures is big, and when the same data are presented in several figures or the data on the same topic is discussed in a different manner in different places (e.g. buoyant density in several figures). As the main results text is very scanty, the results need to be read jointly with the figures. It would be easier for the readers to follow and interpret the data if they were depicted in a few more main figures grouped around the main findings. In my opinion, the main findings are the production (SOP: what and how), characterization (results of rEV properties mentioned in bulk text and then explored and grouped into the supplementary material in detail) and the various examples of use (each with their own dataset with either figure or table). However, this is up to the authors, and they may find a better way to restructure their data.

We agree with the reviewer that the data package is large and that presenting the data in more main figures together with an extended description of the results and discussion could improve the overall flow of the manuscript. So, in agreement with the reviewer's suggestions, we re-structured the manuscript.

The revised manuscript contains 5 main figures:

- 1) Figure 1: rEV are separated from medium conditioned by gag-EGFP transfected HEK293T cells using density gradient centrifugation
- 2) Figure 2: rEV bear physical and biochemical traits characteristic of sample EV
- 3) Figure 3: rEV can be detected and quantified using fluorescence, protein and RNA based assays
- 4) Figure 4: rEV, modified or not by post-insertion PEGylation, act as sample EV when spiked in multiple biofluids
- 5) Figure 5: Use of rEV in various applications

These main figures are supported by 14 supplementary figures that are mostly used to perform additional comparisons using different cell lines, different biofluids, different numbers of rEV spike etc. Experiments performed to study the stability of rEV in different storage conditions are presented in supplementary figures 7-9.

2. Spiking:

There needs to be an explanation and some discussion of how and why the rEV use would allow a significant difference to be detected when it was not measured without the rEV use. It is well known that the plasma of cancer patients has abnormal plasma proteins e.g. acute phase proteins. Could e.g. a corona effect due to the plasma of the cancer patients change the isolation of the rEV and thereby falsely higher values of EVs? (see comment 3 below)

In the proof-of-concept experiment (**figure 5 f**) we demonstrate that rEV can mitigate for intra-assay and inter-user variability while separating EV from plasma samples of healthy donors and cancer patients to reveal true differences in sample EV numbers. Intra-assay variability can be caused by the preparation of SEC columns and density gradients, implementation of products from different batches etc. Inter-user variability can be caused by handling samples over different time points, handling of sample by different operators in similar or different laboratories etc. Indeed, recovery efficiencies of rEV varied between time points and donors as summarized in **supplementary table 2**. This demonstrated that rEV-based data normalization allows to correct for intra-assay and inter-user variability and reveal endogenous sample EV counts/ml biofluid. Especially, some separation efficiencies of plasma from cancer patients were below average, resulting in an increased normalized EV concentration compared to the absolute EV concentration.

Plasma of cancer patients can indeed severely differ from plasma of healthy individuals. As such a higher concentration of immunoglobulins in one or the other group could differentially influence corona formation. However, the proof of concept experiment was performed with proteinase K (PK)-treated plasma, hence clearing rEV membranes from protein corona. Thus theoretically one would expect that a difference in plasma proteins such as immunoglobulins would not influence the results.

Nevertheless, we performed an appropriate control experiment. The figure below presents the density distribution of rEV spiked in PK treated versus not PK treated plasma of healthy individuals and breast cancer patients, measured by fNTA. In the absence of PK treatment (control), the density distribution of rEV differs between healthy individuals and cancer patients. However, PK treatment of plasma, revealed an equal density distribution of rEV for both healthy individuals and cancer patients. This experiment was performed in duplicate (i.e. two different healthy persons and cancer patients). Of note, even between two healthy individuals, a significant variation was observed in the density

distribution of rEV without PK treatment of plasma, as can be seen by the large error bar at 1.141 g/mL.

Was the PEG-rEV used in this study or the unpegylated form? If the unpegylated, please verify the result with the PEG-rEV (see comment 3 below).

In the proof-of-concept experiment we used PK-treated plasma instead of rEV modified by post-insertion PEGylation (rEV-PEG). In the manuscript we demonstrated that the separation efficiency of equilibrium Optiprep density gradient (ODG) centrifugation was not affected when spiking rEV in PK-treated healthy donor plasma versus rEV-PEG in healthy donor plasma (**figure 5 c**). This means that theoretically the normalized number of particles/mL plasma calculated based on the recovery efficiency would be equivalent between PK-treated plasma and rEV-PEG. Since the experiment reported in **figure 5 c** was performed in healthy plasma, we verified these results in cancer plasma (see figure below). This experiment was performed using the same pool of cancer plasma. rEV were spiked in PK-treated plasma (n=3) and rEV-PEG in untreated plasma (n=3). EV were separated from plasma by ODG centrifugation and normalized number of particles/ml plasma were calculated after fNTA quantification. This revealed that the normalized number of particles/mL plasma did not differ between both approaches.

“As an example, sample EV and spiked rEV from plasma of breast cancer patients (n=26) and healthy volunteers (n=11) were isolated by SEC-ODG-SEC, and were measured by NTA and fNTA.” “The average, normalized sample EV concentration per mL plasma was increased by 2.5 fold (5.83x10¹⁰ vs 1.46x10¹¹) (p=0.0002, Mann-Whitney test) in breast cancer patients compared to healthy individuals.” Please use past tense. Please, switch the order of the numbers as now the breast cancer EV amount seems lower than the healthy controls. Was this result measured by NTA or fNTA? Why only one set of results shown, if both methods were used?

We have switched the specified numbers and changed the manuscript into the past tense. In addition, in the revised version of the manuscript we have provided an extended description of the results and figure legends which should clarify which methods have been used for which experiments.

In this specific experiment, we measured the total number of particles with NTA and measured the total number of rEV with fNTA and ELISA. The quantified number of rEV obtained by fNTA and ELISA was used to calculate the recovery efficiency of each separation and to calculate the normalized number of particles/mL plasma (**supplementary table 2**). In **figure 5 f** “absolute” means “the number of particles/mL plasma measured without normalization (only NTA)” and “fNTA” and “ELISA” means “the number particles/mL plasma normalized by fNTA and ELISA respectively (taking into account the recovery efficiency)”.

3. Changes in the buoyant density of the rEV when in plasma:

Since it is crucial for the use of rEV reference as a spike in, the results of the enhanced detection are important, but rather curious, and thus, I would suggest the following controls to be done: Using WB or flow to identify:

- **density position in ODG of plasma-incubated rEVs after PK-treatment (first plasma, then PK)**

This control experiment is presented in supplementary figure 10 b of the revised manuscript. Western blot analysis confirmed that gag-EGFP protein is detected in ODG fractions (1.141-1.186 g/ml) in control plasma compared to ODG fractions (1.086-1.103 g/ml) in PK treated plasma.

- **position in ODG of native plasma EVs after PK-treatment**

This control experiment is presented in supplementary figure 10 e of the revised manuscript. Western blot analysis confirmed that flotillin-1 and syntenin-1 are detected in ODG fractions (1.086-1.103 g/ml) of PK treated plasma. In this experiment plasma was not spiked with rEV.

- **position in ODG of rEVs treated with PK before adding to plasma**

The results of this control experiment are presented in the figure below. Using western blot analysis we first confirmed that rEV are still intact after PK treatment. Indeed, luminal proteins gag-EGFP and syntenin-1 are still intact. By contrast, the tetraspanin membrane protein CD81 was no longer detected using an antibody targeting the extravesicular loop. Next, we established the density distribution of rEV in plasma, treated with different PK concentrations, prior to spiking in plasma. This revealed that pre-treatment of rEV prior to spiking in plasma does not prevent rEV to be shifted to higher density fractions.

These experiments suggest that rEV surface proteins are not required for corona formation. Potentially vesicular lipids attract IgG/IgM and induce corona formation.

4. Patient plasma collection:

Age and sex are known to effect plasma EV concentrations: were the controls matched for age and sex? Of course, 26 is no match for 11, but since age and sex influence, were there males in the controls and was the age similar. If not matched at all, state this in the text as “unmatched controls (sex, age)”.

The healthy controls were matched for sex but not for age. We only included female healthy individuals. The average age of the cancer group was 62.9 whereas the average age of the healthy controls was 48.4. This is because few cancer patients were aged 80 years or older. We clearly stated this in the materials and methods section of the revised manuscript (p23).

Were the healthy controls and the patients treated similarly regarding fasting prior to the blood draw and the time of blood draw?

Cancer patients and healthy individuals were in a non-fasting condition. We checked plasma and eluted SEC fractions visually and observed no milky white colour (indicative of abundant presence of chylomicron). However, proof-of-concept experiments (**figure 5f**) were performed in PK-treated plasma and EV were separated from plasma using equilibrium Optiprep density gradient centrifugation. The orthogonal implementation of size and density based separation (material and methods section p 24) has been reported by our group and others to separate EV with very high specificity from HDL, LDL and chylomicrons^{4,5}.

Was the plasma verified to be platelet-free after double-centrifugation at 15 min 2500 x g? If not, omit the unnecessary and likely false claim of “platelet-free”.

To validate our plasma collection procedure, we checked plasma samples after double centrifugation at 2,500 g with an automated haematology analyser (Sysmex, XP-300) and could not detect any platelets after this procedure. Taking into account the limitations of the haematology analyses (lower detection limit of 1×10^4 platelets/ μL) we consider this plasma as platelet-free.

5. Induced variability:

Abbreviated titles in the table do not explain well enough what was done experimentally and how the calculations based on these results were made. Please, provide more detail of the measurements, procedure and calculations in the main text, or in the M&M section, or in the legend. If 1000 or 500 μL of 2000 μL plasma sample were replaced with PBS, it is 50% not 25%, and 25% not 12.5%, respectively?

We provided a better explanation with regard to the induced variability experiments in the materials and methods section (**p 25**) of the revised manuscript.

We did not replace 1000 or 500 μL of plasma with PBS. Instead we replaced these volumes after ODG-mediated separation of EV from 2 mL plasma, resulting in a 4 mL solution containing sample EV and rEV. So, 1000 μL and 500 μL of a total of 4000 μL are respectively 25% and 12.5%.

6. Were equal number of replicates analysed with each of the methods when the CVs were calculated? Please state the exact number of the replicates for each method in the text.

CV are calculated based upon the graphs presented in **figure 5 a, b**. These experiments were all performed in triplicate, so CV were calculated on the same number of replicates (n=3). We clearly indicated this in the text.

7. How much fluorescence from rEVs was present in the other density fractions (ie in other vesicle types) than those selectively collected for the standard?

In the figure below we present the percentage of rEV quantified in each density fraction using fNTA (left) and the percentage fluorescent particles (ratio fluorescent (fNTA) versus scatter (NTA) measurements) (right). The density fractions collected for rEV preparation are circled in red. They contain the highest % of rEV.

8. If the HEK293T cells (only?) were used for rEV production, it should be stated in the methods. Was the transfection stable or done anew for each isolation? Data in Supplementary Fig 2 could be combined into Supplementary Fig 1, panel b. In my mind, only the repeatability of the isolation (panel c) is relevant information and the rest of the results could be just mentioned in the text. Please, state in the text and M&M from which of the tested cell lines the rEVs for the studies were isolated from. Also, the source of EVs E.g. Figure 1 panel I is missing.

We agree that confusion was induced with regard to the cell line used to produce rEV since the separation of rEV and EV was described in the same paragraph in the material and methods section. In the revised manuscript we have divided this paragraph in “rEV separation” (p21-22) and “EV separation” (p22). For clarity, only HEK293T cells were used for the production of rEV. The other cell lines were used to separate sample EV for the comparison of physical and biochemical characteristics with rEV. HEK293T cells were transiently transfected, meaning that for each new rEV production cells were transfected anew.

As indicated in our answer to comment 1 we combined some panels of figures with some panels of supplementary figures to cover the main findings in 5 main figures. The production of rEV is now presented in figure 1 of the revised manuscript.

In previous figure 1 l (figure 4 a in the revised manuscript), we specified the source of EV spiked in PBS. These were sample EV obtained by ODG centrifugation medium conditioned by MCF7 breast cancer cells.

9. What does this mean: “plasma was diluted with equal volumes(?) of PBS...” Please, clarify in the text if this means 1:1 plasma:PBS. Was equal volume of (PFP)-plasma taken from all study subjects? How much plasma (vol) was then processed for each EV patient/control sample?

This indeed means a 1:1 dilution with PBS. We have now provided this information in the materials and methods section (p23) as well as the volume of plasma used for each experiment.

10. In Figure 1, the number of biological and/or technical replicates is missing in b, c, d, e, f, k, m n, l. Please add this info similarly as it is stated in h-j, unless it is not applicable. Also, please add the relevant method for b (NTA or fNTA). Why is the size distribution of rEV different in Fig 1b from 2b (text 105 nm +/- 9)?

The numbers of biological replicates are now added to the figure legends. In panel b of figure 1 (figure 2 a in revised manuscript) all measurements were performed with NTA. Sample EV cannot be measured with fNTA since these are not fluorescent. The size distribution measurements of rEV, previously presented in figure 1b and 2b, are now combined in **figure 2 a and figure 4 e** in the revised manuscript.

Panel g. Please, show the position of the Fluorosphere bead-based gates and noise. Although not explained, I think this info is partially displayed in Supplementary Fig 9.

The figure below provides the fluorosphere bead-based gates and noise for flow cytometry. This information is also provided in **supplementary figure 14** of the revised manuscript.

Panel n: add PK-treated plasma in the figure.

This is adapted accordingly in figure 4 c of the revised manuscript.

Main Fig 1 electron microscopy images could be larger or with a better resolution: from k it is impossible to see any details. As there is a Supplementary figure 4 for EM images showing practically the same dataset, EM images could be removed from Figure 1 to reduce the data heavy figure and panel k could also be shown larger here. Please rephrase the wording “share morphology” as the morphologies seem to be different in all the EV populations shown. It would also be beneficial for comparison to have the different EVs shown at the same magnification (esp. 4T1).

As we restructured the figures, previous figure 1 k is now presented in a larger format in **supplementary figure 8** (lyophilisation of rEV); EM images for the comparison of the morphology of sample EV and rEV are grouped in figure 2 of the revised manuscript. All EM images are presented at similar magnification. The wording “share morphology” was changed throughout the manuscript to “had similar morphology”, in agreement with the reviewers’ suggestions.

Please show how do the lyophilized and reconstituted rEVs perform in ODG compared to “fresh” rEV? Or could the authors show another, more reliable way for proving intactness of the rEVs post lyophilization? (see comment 17)

To demonstrate that lyophilized and reconstituted rEV remain intact we spiked the reconstituted lyophilized rEV in PK-treated plasma. rEV were then separated by ODG centrifugation from PK-treated plasma and the different density fractions were quantified by fNTA (**supplementary figure 8d in revised manuscript**). This revealed that lyophilized rEV float at EV-characteristic density fractions (1.086-1.119 g/ml).

11. All information of the plasmid pMET7-gag-EGFP is missing from the M&M. By Googling, I found that the plasmid has previously been published by one of the co-authors and that there is a citation

recommended for its use. Please provide the relevant information for this in the text. “pMET7-GAG-EGFP was ... (Addgene plasmid # 80605 ; <http://n2t.net/addgene:80605> ; RRID:Addgene_80605)”. Further, the schematic of the plasmid structure could be placed in Figure 1 a, as it is currently written as to contain this info.

We added the relevant information about the pMET7-gag-EGFP plasmid in the material and methods section (p20). We don't think that the addition of the plasmid information in figure 1 would contribute to the better understanding of the manuscript. Figure 1 in the revised manuscript provides a schematic overview of rEV production indicating cell line (HEK293T) used, amount of cells (3E9) and expected rEV yield (5E11). Detailed protocols on transient transfection, cell culture, conditioned medium preparation etc. are provided in the materials and methods section (p20-22).

12. Supplementary Figure 3:

Why is the data for rEV behavior in ODG given in three separate figures? ODG is indeed an important way to compare the similarity of rEV and native EVs. However, Figure 3 is confusing, because all the different EVs are reported in a dissimilar manner regarding the densities making it very hard to compare. Also, different EV markers are used for the EVs. Please, provide a consistent data set for the same density bins and using the same set of markers. I am particularly concerned about the plasma data. Most likely the EV number/isolation was too low?

Supplementary figure 3 (supplementary figure 2 in the revised manuscript) aims to demonstrate that rEV are floating to equivalent density fractions as sample EV derived from culture medium of other cell lines or from biofluids using equilibrium density gradient centrifugation. Western blot analysis for EV-enriched proteins is an appropriate method to analyse this. We agree with the reviewer that a reorganization of the western blot data and analysis of identical EV-enriched proteins across samples is required to allow for a clear interpretation of the data (supplementary figure 2 in revised manuscript). Therefore, we repeated those experiments while organizing the western blot in a similar manner for all samples. The western blot shown for plasma in the initial supplementary figure 3 was obtained using PK-treated plasma (supplementary figure 7 in revised manuscript). Therefore we also performed an additional separation of sample EV from plasma without PK-treatment. The numbers of EV/mL plasma can differ between plasma samples from different individuals. In addition, separation of sample EV from body fluids is inherently more challenging than separating sample EV from conditioned medium.

13. Supplementary Figure 5: Could the authors comment on the proteins that were found to be unique for rEV or the mock-transfected EVs. Any pathway differences?

Mass spectrometry-based proteomics revealed that rEV contain intraluminal and membrane proteins characteristic of sample EV including ALIX, TSG101, flotillin-1, syntenin-1, CD9, CD81 and CD63. In depth analysis for molecular function and biological process revealed that rEV are enriched in proteins regulating RNA binding and nucleic acid metabolism (**supplementary figure 4 e**). Gag polyprotein contains zinc-finger RNA binding domains and to form VLP, Gag polyprotein must bind RNA^{6,7}. In the absence of viral RNA, gag encapsulates host RNA and any single-stranded nucleic acid longer than ~20–30 nt can support VLP assembly, indicating a general propensity to bind abundant RNA^{8,9}. Indeed, exogenous EGFP mRNA is encapsulated in rEV and allows for a reproducible quantification of rEV using EGFP or gag RT-qPCR assays (**figure 3 j**).

14. Supplementary Fig 6. As the lipid class percentages for rEV were stated in the text, it would be interesting to add in the same parenthesis also the percentages found for the mock- transfected EVs for comparison.

This is indeed an interesting question. The nmol lipid was normalized based on the total amount of protein detected. Since the amount of protein detected per particle was higher for rEV than for the mock EV, the comparison of lipid classes per particles could not be made. For a decent comparison these data should have been normalized for equal amount of particles. This lack of information does not skew the scatter plot data presented in supplementary figure 5a of the revised manuscript because these are relative values of each individual lipid class. If one specific lipid of a lipid class is increased in rEV we observed the same increase in the mock EV.

15. Supplementary Figure 7: Two data sets by different methods are presented, but what is the significance of comparing the different RI of different EVs to this extent? How can the RI information be used? The RI of the most interesting species regarding rEV use as a spike-in (i.e. the plasma EVs) is missing. Please, add. Please rephrase the title (“share refractive index”). As there clearly seems to be challenges using flow cytometry for the RI determination (due to the lower limit of detection? Also, the x axis of panel B does not show well, please state values for comparison in the fig., if possible), could the dataset in Fig 7 be reduced by omission of the flow data?

rEV have a refractive index (RI), a physical property determining the amount of light that is scattered by a certain material, equivalent to sample EV, indicating that rEV can be used as a calibrator for measurement instruments. To compare or reproduce EV research, exact enumeration of EV particles rather than indirect quantification via protein concentration measurements is recommended¹⁰. Flow cytometry and NTA are frequently used to quantify individual particles. Synthetic silica or polystyrene

beads with a RI of respectively 1.45 and 1.61 are commonly used to calibrate these instruments. Since the RI determines the minimal size of particles detected with NTA and determines the relationship between scatter and size in flow cytometry, the average size and concentration of sample EV detected by these methods is underestimated^{11,12} (discussion section of revised manuscript p16).

As requested, we performed an additional analysis to identify RI of plasma EV by the implementation of NTA and MIE theory (**supplementary figure 3** in revised manuscript). Furthermore we omitted RI results obtained by Flow-SR. In conclusion, we have shown that the RI of rEV is not higher than that of sample EV, namely that it is lower than 1.40.

Not being an expert, I would like to know does the fluorescence from EGFP in rEV (up to 5000 copies in VLP according to introduction) influence the determined RI? Or is EGFP not excited at the relevant wavelengths in the used methods? Please, comment.

Because a fluorescent particle absorbs light, fluorescence affects the optical properties of a particle, as suggested by the reviewer. To account for absorption, the physical property “refractive index” is divided into a real and an imaginary part. The real part of the refractive index describes the phase velocity of light in the material. A contrast in the real refractive index (and in the velocity of light), for example between the medium and a vesicle, causes light scattering, as measured by NTA and flow cytometry. Thus, where we state refractive index in the manuscript, the reviewer should interpret this as the real part of the refractive index.

The imaginary part of the refractive index describes optical losses in a material, for example due to the absorption of light by fluorescent molecules. We have performed Mie calculations to show that for submicrometer particles the real part of the refractive index dominates the scattering properties of a particle. The reason is that, compared to scattering, absorption of fluorescent molecules is negligible for the length scale of vesicles.

The number of biological and technical repeats is missing.

Mie theory describes light scattering of spheres of all size parameters. The model used representative NTA trajectories of rEV and sample EV from various sources. The mathematical calculation has been performed once using one representative NTA trajectory from each of the sample EV obtained from various sources (MCF7, 4T1, CAF, urine, plasma). With these results we show that the RI of rEV, calculated by Mie theory, was not higher than 1.40, this is in contrast to other reference materials (beads,...) used to calibrate measurement instruments in EV research.

What is the relevance for Supplementary Figure 14? Could this be replaced with a reference or combined with Supplementary Fig 7?

We now combined this supplementary figure 14 with the dataset of the RI of the different EV in supplementary figure 3 in the revised manuscript.

16. Supplementary Figure 8 panel D. Please increase the font size for the rEV concentrations in the figure. How was the concentration measured? fNTA or NTA?

The font sizes have been increased. To establish linear or semi-logarithmic regression curves for fluorescent microplate reader, western blot, ELISA and qPCR analysis, rEV were quantified by fNTA (results section p8 in the revised manuscript).

17. Supplementary Fig 9: Please also provide the one year storage data, or explain better in the legend when FT1 and 2 measurements were done. Could panel D be provided as overlapping plots to ease comparison? Alternatively, add the mean and the mode information in the figure. Panel e: Please provide the event counts before and after lyophilization in the figures. It seems that there are more counts after lyophilization in the detectable range, which could indicate break-up/fusion? Please, discuss. See also comment 10.

FT1 and FT2 in supplementary figure 9 included one year storage data. These data are presented in **supplementary figure 7** of the revised manuscript. We have changed FT1 and FT2 to 6 months and 12 months in the corresponding figure legend.

Supplementary figure 9 d is **supplementary figure 8 a** in the revised manuscript. We did not provide overlapping plots but increased the font size of the modi of the peaks for the ease of comparison.

The event counts are now added in the scatterplots of rEV before and after lyophilisation; the event count was not significantly different after lyophilisation (**supplementary figure 8b** in the revised manuscript). When comparing the scatter plots of rEV before and after lyophilization, we observed that the majority of lyophilized rEV localize to the same position in the plots compared to fresh rEV. However, a small percentage (+/- 2.5%) of rEV in the lyophilized sample displayed higher light scattering levels. The GFP levels in this population were not increased, so the effect was most likely not due to clumping/aggregation of rEV. These higher light scattering levels could be a consequence of the addition of trehalose to the rEV suspension followed by incomplete rehydration.

18. P-values: I would find reporting the nonsignificant values in the case of negative results (e.g. Fig 9a-c storage data, SFig 2b viability data etc) useful.

Where appropriate, we added these non-significant p-values to the manuscript.

Technical comments (minor)

1) How was the freezing of samples/rEV performed? (snap or?). How was thawing performed? Please, add this information.

Samples were placed in a tube rack in an ice bucket and transferred to the -80 °C freezer for long term storage. rEV were stored in 10 µl aliquots which were thawed by holding the tube by hand and monitoring thawing. This information is now also added in the materials and methods section (p29).

2) “Efficiencies of SEC, two-dimensional size- and density-based separation (SEC-ODG-SEC)²⁰, and differential ultracentrifugation (dUC) to isolate sample EV from plasma were respectively nearly 100%, 30% and 10% (figure 2 e, f).” Should this be rEV? Also, once something is abbreviated (SEC-ODG-SEC, please use the abbreviation also later in the text.

The separation efficiencies were calculated for sample EV based upon the recovery of rEV. The term sample EV was correctly indicated in the text. rEV is fully characterized and is commutable during EV separation from plasma. Therefore, by using rEV to calculate the separation efficiency of a certain method, we could extrapolate these results to sample EV being separated from plasma.

We have checked the abbreviations and used them throughout the manuscript.

3) For the following assays data concerning the sample numbers/sample size/used material (volume/particle number/ protein) for each assay would be important information:

Protein mass spectrometry

Lipid mass spectrometry

Total cholesterol

ELISA

Related comment: Please elaborate in the text what this means: “To quantify the total amount of phospholipids, we summed the abundances of individually measured species within each phospholipid class and normalized based on the amount of protein.”

We added this information in the materials and methods section or figure legends. Since we omitted the data concerning the total amount of phospholipids and only included the relative values, we omitted this as well from the M&M section.

4) Immuno-EM: Please add info of the used dilution of primary antibody (anti-CD63) and the secondary. Gold size, dilution and the commercial source for protein A are also missing. Fig 1 legend is misleading, because it describes the use of a gold-conjugated antibody, which is not the case according to M&M?

We added this information in the materials and methods section (p19 in revised manuscript).

5) Supplementary table 2: please add the word EV where appropriate.

Adaptions have been made to this table as suggested (supplementary table 3 in revised manuscript).

Content and writing (major)

1. The abstract states that a SOP will be communicated in the ms. However, after reading the main text & figures, it is unclear for me what is the SOP and what is it provided for? Please clarify in the text (aim & discussion), and also e.g. by mentioning in Supplementary Figure 1, if that describes the intended SOP for rEV production? Regarding the SOP, it should also be clearly defined in the M&M as now the production of rEV and isolation of EVs from different sources run together.

The SOP mentioned in the abstract was indeed referring to the production and use of rEV. To avoid misinterpretation, we omitted the term "SOP" in the abstract. Nevertheless, the materials and methods section contains all required information to reproduce the experiments including the production and characterization of rEV.

2. The manuscript is clearly and fluently written, but it has a large amount of grammatical errors regarding e.g. missing punctuation (participial phrases such as "To understand and mitigate these limitations, we have..."). Most importantly, the use of the present tense when describing results of own data leads to difficulties in separating pre-existing information, accepted knowledge and facts from the results generated in the experiments of this study. I strongly urge changing the verbs in the description of any experimentation into past tense. E.g. in the abstract "Implementation of rEV reduced intra-method and inter-user variability of EV sample preparation and analysis, and improved the sensitivity of EV enumeration in biofluids." states an observation, whereas in the present tense, as it is now in the text, it sounds like a statement or fact.

We thank the reviewer for this suggestion and we have adapted all observations from experiments to the past tense. We also tried to resolve any missing punctuation in the revised manuscript.

3. The use of “commutable” in the abstract and at several places in the text as an adjective is not easily understood for the indicated meaning of the rEVs’ analogous behavior with native EVs under various experimental conditions. Please rephrase or -word.

We rephrased the adjective “commutable” in the text to “behave similar as sample EV under various experimental conditions”. We also gave some extra information about the word “commutability” and explained better its importance (p4 and p11).

4. The use of word “share” with morphology/ size /density etc. “Had similar” or “did not differ” would be more appropriate.

We replaced the word “share” to “had similar” according to the reviewers suggestions.

Minor comments

What does the informed use of rEV mean? Informed by whom? Could the authors elaborate their meaning or remove.

We mean that the researcher should not blindly use rEV but also know what he/she is doing and why. For clarity we have removed “informed use” from the manuscript text.

What does “accurately measured” mean? Wouldn’t accuracy require a standard, which is currently being developed in the study? Please rephrase or remove.

We agree with the comment from the reviewer. We have omitted accurately in the manuscript text.

behaved in a linear manner?: “rEV behave linear in defined concentration ranges when diluted in PBS and measured by a fluorescence microplate reader, densitometry of EGFP western blot bands and an ELISA for p24, a subunit of the gag polyprotein”

We rephrased this in the manuscript text.

was/could?:

The latter being the most sensitive as it can detect rEV in a range of 10⁶-10⁷ rEV

This is adapted in the text.

What is size fraction?: rEV eluted in the same size fractions as sample EV upon isolation from cell culture medium, urine and plasma using size exclusion chromatography (SEC)

Here we mean the fractions eluting from a SEC column. As this could be confusing we changed this in the manuscript to “elution volumes”.

What does inducible mean here? :“This shift was reversible and inducible.”

With inducible we mean that this shift can be experimentally induced in biofluids or buffers, where it is normally not occurring, by the addition of physiological amounts of immunoglobulins. This is now explained in the manuscript text.

“Treatment of what? with proteinase K (PK) before gradient separation reverses the density shift...”

Plasma or rEV?

“Treatment of concentrated SEC fractions before ODG centrifugation”. This is now explained in the manuscript text.

Please, de-abbreviate EGFP at the first mention. Please, also check the abbreviated material throughout the manuscript, i.e. do not abbreviate if you do not use the abbreviation.

EGFP has now been de-abbreviated at its first mention. We also checked other abbreviations used throughout the manuscript.

5. References:

• **“Current reference materials for calibration such as polystyrene beads, silica beads or liposomes lack EV-like properties, resulting in an aberrant refractive index (RI) and inaccurate measurements^{7,8}.” Do these references cover liposomes? If not, remove.**

Yes. The Valkönen paper reviews the advantages and disadvantages of liposomes as reference material for EV.

• **“However, EV research and its biomedical applications are hampered by the myriad of isolation and measurement methods and the lack of appropriate reference materials for accurate calibration, normalization and method development^{4–6}.” Please, change ref 5 to the newest MISEV guidelines**

They & Witver JEV2018, where the most recent understanding of the complexity of the EV field regarding isolation, lack of reference materials, etc is addressed.

We have changed the reference from MISEV2014 guidelines to the updated MISEV2018 guidelines.

- **“The suitability of HIV-1 VLP as a reference material for EV research has not been previously considered or tested.” This is true and important novelty of the study. However, although I am not an expert, retroviral protein-aided transfer of protein has been previously described as a strategy for other purposes with EVs e.g. vaccination platforms in the literature/patents. Discussion and referencing of this in regard to the development in the field would be nice. I leave the choice of the examples to the discretion of the authors.**

We now included a separate part in the discussion describing the possible use of VLP for the transfer of active protein and RNA to other cells (page 15, reference 30).

- **Patent search by the indicated patent number did not produce any results. Please, verify the number.**

The number is correct; however, the patent is not yet public. We have submitted the patent November 9th, 2017. Evaluation by the European patent office resulted in a positive search report, meaning that the patent will probably be approved in the second round. Typically, a patent is published 18 months after initial submission, so we expect the publication of our patent by the European patent office around May 2019. We will update the status during the reviewing/editorial process.

Reviewer 3

1. The rEV was produced from transiently transfected HEK293T cells. Thus, it is hard to believe that all the cells produce rEVs. This indicates that there should be the heterogeneous mixture of EVs (rEVs and EVs) in the culture medium. Even in the transfected cells, rEV production abilities among cells are different that native EVs as well as rEVs can be also produced at the same time. Some of the analytical methods used in this research cannot discriminate rEVs from EVs, there is a lack of evidence for most of the analytical data comparing rEVs with sample EVs, which significantly weakens authors' conclusion.

In supplementary figure 8a of the initial submission we showed that on average 80 % of all particles detected in rEV preparations using NTA are also detected with fNTA. As such, we agree with the reviewer that not all particles in rEV preparations obtained by equilibrium Optiprep density gradient (ODG) centrifugation of medium conditioned by gag-EGFP transfected HEK293T cells are fluorescent. One possibility, as suggested by the reviewer, is that rEV preparations are contaminated with approximately 20% of EV originating from endogenous EV biogenesis pathways of HEK293T cells.

In response to reviewer 1 (comment 1 and 2) we performed Optiprep velocity gradient (OVG) centrifugation according to the protocol described by Dettenhofer et al¹. Medium conditioned by gag-EGFP transfected HEK293T cells was separated by OVG centrifugation. Different OVG fractions collected from the top to the bottom were analysed by western blot analysis and nanoparticle tracking analysis in scatter mode and fluorescence mode. Western blot analysis revealed that gag-EGFP positive rEV in OVG fractions 10-13 are efficiently separated from gag-EGFP negative endogenous EV in OVG fractions 4-7 (**supplementary figure 6 a**). Nanoparticle tracking analysis identified that rEV separation from medium conditioned by HEK293T cells by OVG centrifugation resulted in approximately 100% fluorescent particles (**supplementary 6 c**).

In addition, we have extensively analysed the physical and biochemical characteristics of rEV obtained by OVG centrifugation. As for rEV obtained by ODG centrifugation and sample EV, rEV obtained by OVG centrifugation showed a characteristic cup-like morphology by TEM analysis (**supplementary figure 6 e**), contained all frequently assessed EV-enriched proteins (including Alix, TSG101, flotillin-1, CD81 and CD9) as shown by western blot and mass spectrometry based proteomics (**supplementary figure 6 b, j**) and floated to the characteristic buoyant density of 1.086-1.119 g/ml when subjected to equilibrium based ODG centrifugation. Also, rEV separated by OVG centrifugation from medium conditioned by gag-EGFP transfected HEK293T cells were detectable in a linear manner using a

fluorescence plate reader and the p24 ELISA and correlated semi-logarithmically with Cq values for EGFP mRNA targeted RT-qPCR (**supplementary figure 6 f, g, h**).

In conclusion, the Optiprep velocity density gradient enables the separation of rEV from the conditioned medium of gag-EGFP transfected HEK293T cells and results in a nearly 100% fluorescent gag-EGFP positive rEV population. As indicated, all these data are presented in the manuscript as supplementary figure 6 and are described in results section p 8-9 and in discussion section p 15.

2. Although there are results showing rEV can be used as a reference to EV, there is no direct evidence showing why rEV is superior to other reference materials.

3. In addition, there are lots of research articles about the use of exogenously or endogenously labeled EVs with fluorescent molecules. What will be the potential issues for using them as reference materials? In this context, what is the novelty of the rEV?

The urge for a biological reference material for EV research and biomedical applications is increasingly recognized^{10,13,14}. To meet this need we have developed rEV, a gag-induced mimic of EV. rEV are generated by reproducible transfection of HEK293T cells with gag-EGFP DNA followed by high-purity rEV separation from CM using density gradient centrifugation¹⁵. rEV are safe since the viral genome, proteases and other viral proteins which render viruses infectious are not present¹⁶. rEV show biochemical and physical traits characteristic of sample EV¹⁷⁻²¹: 1) equivalent lipid and protein profiles including the presence of intraluminal and membrane EV-associated proteins; and 2) equivalent cup-shaped morphology, size (50-180 nm), buoyant density (1.086-1.119 g/mL), RI (1.37) and zeta potential (-32 mV). rEV are intrinsically stable during lyophilization and long-term storage, ensuring convenient distribution and use. They contain exogenous mRNA and protein allowing quantification with multiple measurement instruments while ensuring adequate differentiation from sample EV. OVG centrifugation to separate rEV from medium conditioned by HEK293T cells results in rEV preparations with high specificity; indeed, approximately 100% of particles detected in the preparation are fluorescent and thus contain gag-EGFP protein (**supplementary figure 6 and answer to comment 1 reviewer 1 and reviewer 3**). rEV are the first biological reference material for EV compatible with a plethora of applications including quality control, instrument calibration and data normalization (**figure 5g**).

We are aware of other biological reference materials including commercially available reference materials retrieved from different biofluids by differential ultracentrifugation (HansaBiomed) and fluorescent labelling of sample EV (i.e. endogenous EV) by fusing tetraspanins with fluorescent

proteins, lyphophilic dye labelling of EV membranes or targeting EV surface proteins with fluorescent tagged antibodies. We have tested and compared the performance of rEV with those potential reference materials based upon fluorescence intensity using fluorescent nanoparticle tracking analysis (fNTA) in scatter and fluorescence mode (**see figure below and supplementary figure 12 of revised manuscript**).

a) and b) in the figure below represent size distributions obtained by NTA measurements in scatter and fluorescence mode of respectively fluorescently labelled EV separated by differential ultracentrifugation from medium conditioned by HCT-116 colorectal cancer cells and plasma that are commercialized by Hansabiomed (http://www.exotest.eu/online_orders/image/data/inserts%20sheets/FluorescentExosome%20standards.pdf). Differential ultracentrifugation is known to be prone to co-sedimentation of soluble protein, protein aggregates and lipoproteins^{10,15,22}. We confirmed this by ponceau red staining of protein bands on a western blot membrane containing proteins from identical numbers of rEV (separated by ODG centrifugation) and reference EV particles from Hansabiomed (separated by differential ultracentrifugation), which clearly revealed abundant presence of proteins including presumably the typical albumin band at 65 kDA (**represented in f**). Nevertheless, Hansabiomed provides reference EV in µg of protein rather than particles/mL. Scatter-based NTA analysis revealed that these reference EV have a very irregular size distribution, independent from the source origin (cell culture or plasma), compared to rEV (**represented in e**). In addition, fluorescence-based fNTA analysis was not able to detect any fluorescent particle above the lower detection limit of the instrument (1E8 particles/ml) resulting in an irregular size distribution; this is sharp contrast to rEV revealing approximately 100% fluorescent particles using fNTA (**represented in e**).

c) Represents size distributions obtained by NTA measurements in scatter and fluorescence mode of CD63-EGFP labelled EV kindly provided by professor Bernd Giebel. Alike Hansabiomed, those EV are obtained by differential ultracentrifugation²³. By fNTA we could only detect 2.5% fluorescent particles indicating that CD63-EGFP labelled reference EV are inferior to rEV (represented in e, approximately 100% fluorescent particles). fNTA analysis of CD63-EGFP labelled EV was also performed by Gorgens et al. and reported to contain 16% fluorescent particles. This is potentially due to the combination of the lower number of CD63 protein per EV (compared to gag) resulting in a reduced fluorescence intensity and/or the presence of CD63 on selective EV subpopulations resulting in a partial labelling of some EV but not all EV.

d) Represents size distributions obtained by NTA measurements in scatter and fluorescence mode of palm-EGFP EV separated by ODG centrifugation from medium conditioned by HEK293T cells

transfected with EGFP coupled to a palmitoylation signal. This palmitoylation signal targets EGFP to the inner side of the cell membrane, hence also fluorescently labelling EV generated by different biogenesis pathways from these cells²⁴. By fNTA we could only detect 13.2% fluorescent particles indicating that rEV is also superior to this potential reference material (represented in e, approximately 100% fluorescent particles).

Finally, lyphophilic dye (PKH67) labelled EV are sensitive to photo bleaching hampering their analysis by fNTA.

Important to note is that all reference materials were assessed by NTA using the same instrument settings (**see materials and methods section p 25-26**).

Besides the observation that rEV outperforms the above tested potential reference alternatives (Hansabiomed, CD63-EGFP, palm-EGFP and PKH67), rEV provides a plethora of applications. rEV can be quantified and distinguished based on the presence of a non-human protein (gag and EGFP), but also a non-human RNA (EGFP mRNA and gag mRNA); a feature not provided by the tested reference alternatives.

We have added the performance comparison of rEV versus potential reference EV alternatives to the discussion section p15-16 and visualized the results in **supplementary figure 12** of the revised manuscript.

4. There is no information on the amount of spiked rEV into each solution. In supplementary table 1, it is only mentioned that $1E^{10}$ rEV is spiked into plasma (to what volume?).

The amount of spiked rEVs and the ratio between spiked rEVs and already present EVs are important especially in the downstream analysis of EV biomolecules including proteins, RNAs, DNAs, etc. This is because of rEVs are also produced from the cells that they contain most of the biomolecules that sample EVs have. Thus, if there are too many rEVs spiked into the samples as compared to already present EVs in the sample, some of the specific biomolecules from sample EVs can be over- or under-estimated. This will critically limit the use of rEV as a reference material in the biomedical fields. In addition, the concentrations of sample EVs are entirely different among the bodily fluids including plasma and urine. So, for the practical use of rEV, the effect of amount and ratio of rEVs should be carefully studied to ensure its use as a reference material in this field.

We agree with the reviewer that it is important to specify the spiked amount of rEV for each individual experiment. We have now provided this information in the revised manuscript in both the results section, figure legends and material and methods section.

We understand the issue raised by the reviewer concerning contamination of sample EV with RNA and proteins derived from spiked rEV when performing downstream omics analysis and by extension also functional analysis. That is exactly the reason why, in the manuscript, we also present a modification to rEV that allows to separate rEV from sample EV. This modification consists of post-insertion PEGylation of rEV (rEV-PEG). We have demonstrated that rEV-PEG are commutable and allow for an efficient separation from sample EV using anti-PEG coated magnetic beads or affinity columns coated with these antibodies (**figure 4 d-g and results section p11-12**). This approach allows researchers to first calculate the separation efficiency, then to capture and segregate rEV-PEG from sample EV and finally to perform downstream experiments using sample EV (i.e. proteomics and RNA sequencing).

The normalized total number of EV detected in plasma of healthy persons and cancer patients is summarized in Supplementary table 2. We detected on average $1.5E11$ and $3.3E11$ EV in 2 mL plasma of respectively healthy persons and cancer patients as quantified by NTA. For those experiments we spiked $2E10$ rEV, meaning that we spiked around 10% of the initial number of sample EV present in 2 ml of plasma. The number of spiked rEV appropriate for data normalization depends on the recovery efficiency of the separation method and the sensitivity of the rEV detection method. Taking into account the separation efficiency of ODG ultracentrifugation to separate EV from plasma (30% as reported in figure 5 a, b and c) and the lower detection limit of fNTA (with a linear detection range of $1E8$ - $1E9$ rEV as reported in figure 3 c) a $2E10$ rEV spike was required. Analysis of recovery efficiency by p24 ELISA assay, the most sensitive detection assays

with a linear detection range of $1E6$ - $1E7$ rEV would allow to spike a 100-fold lower amount of rEV ($2E8$ rEV or 0,1% of the initial number of sample EV present in 2 ml of plasma). To calculate the endogenous number of sample EV in urine, we spiked urine samples (10 ml) from six healthy donors with $2E10$ rEV and calculated the recovery by fNTA (**see figure a below**). The normalized EV/ml urine ranges between 6 - $9E10$ EV resulting in 6 - $9E11$ total sample EV in 10 ml of urine. Thus, we spiked approximately 3% of the initial number of sample EV present in 10 ml of urine; 100x less rEV could be spiked if ELISA instead of fNTA is used as detection method.

To address the concern of the reviewer with regard to impact of the relative number of rEV spiked compared to the number of sample EV present in the biofluid, we spiked different numbers of rEV in plasma and calculated the separation efficiencies of ODG centrifugation and SEC using respectively fNTA and ELISA. We observed no significant differences ($p > 0.0333$, Mann-Whitney test) in the recovery efficiencies obtained with different amounts of rEV spike. The results of this experiment are visualized below (b and c) and in **supplementary figure 11 of the revised manuscript**.

References

1. Dettenhofer, M. & Yu, X. F. Highly purified human immunodeficiency virus type 1 reveals a virtual absence of Vif in virions. *J. Virol.* **73**, 1460–1467 (1999).
2. Cervera, L. *et al.* Generation of HIV-1 Gag VLPs by transient transfection of HEK 293 suspension cell cultures using an optimized animal-derived component free medium. *J. Biotechnol.* **166**, 152–165 (2013).
3. Briggs, J. A. G. *et al.* The stoichiometry of Gag protein in HIV-1. *Nat. Struct. Mol. Biol.* **11**, 672–5 (2004).
4. Tulkens, J. *et al.* Increased levels of systemic LPS-positive bacterial extracellular vesicles in patients with intestinal barrier dysfunction. *Gut* gutjnl-2018-317726 (2018). doi:10.1136/gutjnl-2018-317726
5. Onódi, Z. *et al.* Isolation of High-Purity Extracellular Vesicles by the Combination of Iodixanol Density Gradient Ultracentrifugation and Bind-Elute Chromatography From Blood Plasma. *Front. Physiol.* **9**, 1479 (2018).
6. Comas-Garcia, M. *et al.* Dissection of specific binding of HIV-1 Gag to the ‘packaging signal’ in viral RNA. *Elife* **6**, (2017).
7. Rulli, S. J. *et al.* Selective and nonselective packaging of cellular RNAs in retrovirus particles. *J. Virol.* **81**, 6623–31 (2007).
8. Campbell, S. & Alan, R. In Vitro Assembly Properties of Human Immunodeficiency Virus Type 1 Gag Protein Lacking the p6 Domain. *J. Virol.* **67**, 5550–5561 (1999).
9. Pastuzyn, E. D. *et al.* The Neuronal Gene Arc Encodes a Repurposed Retrotransposon Gag Protein that Mediates Intercellular RNA Transfer. *Cell* **172**, 275–288.e18 (2018).
10. Théry, C. *et al.* Minimal information for studies of extracellular vesicles 2018 (MISEV2018): a position statement of the International Society for Extracellular Vesicles and update of the MISEV2014 guidelines. *J. Extracell. Vesicles* **7**, 1535750 (2018).
11. Valkonen, S. *et al.* Biological reference materials for extracellular vesicle studies. *Eur. J. Pharm. Sci.* **98**, 4–16 (2017).
12. Van Der Pol, E., Coumans, F. A. W., Sturk, A., Nieuwland, R. & Van Leeuwen, T. G. Refractive index determination of nanoparticles in suspension using nanoparticle tracking analysis. *Nano Lett.* **14**, 6195–6201 (2014).
13. Gardiner, C., Ferreira, Y. J., Dragovic, R. A., Redman, C. W. G. & Sargent, I. L. Extracellular vesicle sizing and enumeration by nanoparticle tracking analysis. *Journal of Extracellular Vesicles* **2**, (2013).

14. De Wever, O. & Hendrix, A. A supporting ecosystem to mature extracellular vesicles into clinical application. *EMBO J.* 1–5 (2019). doi:DOI 10.15252/emboj.2018101412 (In press).
15. Van Deun, J. *et al.* The Impact of Disparate Isolation Methods for Extracellular Vesicles on Downstream RNA Profiling. *J. Extracell. vesicles* (2014).
16. Zhao, C., Ao, Z. & Yao, X. Current Advances in Virus-Like Particles as a Vaccination Approach against HIV Infection. *Vaccines* **4**, (2016).
17. Booth, A. M. *et al.* Exosomes and HIV Gag bud from endosome-like domains of the T cell plasma membrane. *J. Cell Biol.* **172**, 923–35 (2006).
18. Gould, S. J., Booth, A. M. & Hildreth, J. E. K. The Trojan exosome hypothesis. *Proc. Natl. Acad. Sci. U. S. A.* **100**, 10592–7 (2003).
19. Cashikar, A. G. *et al.* Structure of cellular ESCRT-III spirals and their relationship to HIV budding. *Elife* **3**, (2014).
20. Bieniasz, P. D. Late budding domains and host proteins in enveloped virus release. *Virology* **344**, 55–63 (2006).
21. Jouvenet, N. *et al.* Plasma membrane is the site of productive HIV-1 particle assembly. *PLoS Biol.* **4**, e435 (2006).
22. Coumans, F. A. W. *et al.* Methodological Guidelines to Study Extracellular Vesicles. *Circ. Res.* **120**, 1632–1648 (2017).
23. Görgens, A. *et al.* Optimisation of imaging flow cytometry for the analysis of single extracellular vesicles by using fluorescence-tagged vesicles as biological reference material. *J. Extracell. Vesicles* **8**, 1587567 (2019).
24. Lai, C. P. *et al.* Visualization and tracking of tumour extracellular vesicle delivery and RNA translation using multiplexed reporters. *Nat. Commun.* **6**, 7029 (2015).

REVIEWERS' COMMENTS:

Reviewer #1 (Remarks to the Author):

The authors have responded appropriately to my suggestions, and I have not further changes to suggest...with the minor exception of copy-editing of "OptiPrep," which is incorrectly capitalized several times.

Reviewer #2 (Remarks to the Author):

The manuscript of Geurickx et al (NCOMMS-18-36122-T) has been considerably improved upon revision and by the answers and modifications addressing the reviewers' concerns or suggestions for amendments.

There are still few omissions and unclarities in the text, the addressing of which would improve the readability and cohesion of the paper before its publication.

Technical (major):

1. M&M: Please provide the information of how much plasma was used in the patient vs healthy control experiments/donor.
2. Please add in the Materials and Methods which densities were collected from the ODG gradient and how many fractions were pooled.
It is mentioned in the text that densities 1.086-1.119 were collected for the rEV. However, the provided data shows that rEV in plasma are contained in fractions with densities of 1.141 and 1.186. Please explain/remedy.
3. Please check all source files for completeness of data. I could not locate from the source files e.g. the electron micrographs for Supplementary Figure 6.

Technical (minor):

1. p.7 TEM: Please remove the comments regarding "phospholipid bilayer" from the results here and also later (p.9). To accurately verify a lipid bilayer, you need cryo-EM or TEM sections.
2. p.7 Assessment of EV markers: immune-EM was only performed to show the presence of CD63? Currently, the sentence is misleading as it states that ALIX, Tsg 101, flotillin-1 etc were shown by WB, immune-EM and proteomics. Please correct. As the proteomics data is not yet available for checking in PRIDE, I assume that the stated proteins can be found there upon data release.
3. Please write out the result for Figure 4e in the Results section.
4. Please add (3h) at an appropriate place in text. The order of ELISA and gag-EGFP are not the same in the Results text and in Fig 3.
5. "Freezing (for 6 months and 12 months) and thawing of rEV (supplementary figure) followed by storage of rEV at 4C..."
Surely the freeze-thaw experiments and +4C storage experiments were not connected? Please rephrase.
Figure still contains FT1 which in another figure means Flow Through, but here it is a remnant of Freeze-Thaw1 in the previous version?
6. This previous query was not addressed: For the following assays data concerning the sample numbers/sample size/used material (volume/particle number/ protein) for each assay would be important information:
Protein mass spectrometry
Lipid mass spectrometry
Total cholesterol
ELISA
(Number of repeats (technical) or biological replicate is still missing at least for lipidomics.)

Content & writing (major):

1. Introduction:

- a. behave *similarly*
- b. Current reference materials...liposomes, *which* lack EV-like properties?
- c. ...are commercially available, but *they* lack...?
- d. Recent approaches *have* tried...?
- e. rEV **find their origin** in the major structural component of HIV-1 virus particles...?

2. Results: The last sentence of the first para (*"We evaluated at least three biological replicates of rEV... "*) seems to be out of place here, since by 1f nothing has been done to rEV yet as comparisons, which start from Fig 2. Reposition?

3.

- a. Supplementary Fig 2 data addressed before the real Fig 2? Change order?
- b. p.10 *exclusion chromatography* is missing from line 13.
- c. p.11 Suggestion (as plasma behaves differently and is described separately after the other biofluids): replace **various** with *these*.
- d. The sentence *"This shift was reversible and inducible"* is still in text, although the authors claim to have corrected it. **Suggestion**: *"This shift was reversible, as proteinase K (PK)... "*
At the end of the paragraph: *"This shift was inducible, as an addition of physiological amounts of IgM..."*

4. Why is there on p.12 a reference (19) for ODG centrifugation, but no reference for SEC or differential centrifugation?

5. Discussion:

- a. *An urge to obtain* a biological reference material or *A need for* a biological ...?
- b. To meet this need,
- c. a gag-*protein -induced*
- d. rEV were generated...
- e. rEV are safe, since
- f. rEV are **intrinsically stable**... What does intrinsically stable mean?
- g. In depth analysis for molecular function and biological *processes* revealed...
- h. What does this mean: **"As such, rEV-PEG do not interfere with downstream analysis of sample EV composition (proteomics, lipidomics, transcriptomics, metabolomics) and function."** Please, rephrase.
- i. **(size and large EV-specific protein versus other)**? Other what?

6. Figures:

- a. In Fig 1 the schematic of rEV is very useful. However, as it highlights only some of the sample EV-related molecules shared with the rEV (although many more were discovered by proteomics and lipidomics), it may be helpful for the reader if the exemplary nature of the depicted molecules is somehow stated e.g. "Schematic presentation of rEV showing representative/some of the key/molecular components shared with sample EVs..."
- b. Supplementary Fig 1d: n=6 refers to technical or biological replicates?

7. References:

- a. Figure 1. The reviewer fully understands the need to explain the biogenesis of rEV in panel b and is convinced that the readers will also appreciate it. However, since none of the information in panel b was generated in this study, it would be appropriate to reference the source for the information.

Reviewer #3 (Remarks to the Author):

Since the authors made all the responses to my comments, the manuscript seems good to be published.

Responses to the referees' comments

Reviewer 1

The authors have responded appropriately to my suggestions, and I have not further changes to suggest...with the minor exception of copy-editing of "OptiPrep," which is incorrectly capitalized several times.

We thank the reviewer again for the valuable suggestions that were given.

We checked the manuscript and Capitalized OptiPrep correctly throughout the manuscript.

Reviewer 2

The manuscript of Geurickx et al (NCOMMS-18-36122-T) has been considerably improved upon revision and by the answers and modifications addressing the reviewers' concerns or suggestions for amendments.

There are still few omissions and unclarities in the text, the addressing of which would improve the readability and cohesion of the paper before its publication.

Technical (major):

1. M&M: Please provide the information of how much plasma was used in the patient vs healthy control experiments/donor.

We provided the total volume of plasma used for the proof of concept experiment in the results section (**p 13**). The adaptation was made in track changes.

2. Please add in the Materials and Methods which densities were collected from the ODG gradient and how many fractions were pooled.

It is mentioned in the text that densities 1.086-1.119 were collected for the rEV. However, the provided data shows that rEV in plasma are contained in fractions with densities of 1.141 and 1.186. Please explain/remedy.

We provided the correct densities and the amount of collected fractions.

The reviewer is correct concerning the fact that rEV is collected in fractions 1.086-1.119 from CCM and that they are present at higher densities when spiked in plasma. We do however explain very well in the text that this shift is occurring in plasma and that it can be reversed by PK treatment or PEGylation of rEV and that it can be induced by the addition of IgG and IgM (p11-12).

3. Please check all source files for completeness of data. I could not locate from the source files e.g. the electron micrographs for Supplementary Figure 6.

All source data files are checked and relevant data were added where they were missing.

Technical (minor):

1. p.7 TEM: Please remove the comments regarding “phospholipid bilayer” from the results here and also later (p.9). To accurately verify a lipid bilayer, you need cryo-EM or TEM sections.

We removed this statement in track changes.

2. p.7 Assessment of EV markers: immune-EM was only performed to show the presence of CD63? Currently, the sentence is misleading as it states that ALIX, Tsg 101, flotillin-1 etc were shown by WB, immune-EM and proteomics. Please correct. As the proteomics data is not yet available for checking in PRIDE, I assume that the stated proteins can be found there upon data release.

We followed the reviewers suggestions and separated the claim of CD63 presence from the western blot results and only claim it through proteomics and IEM. All changes were made in track changes.

The proteomic data will be available upon publication. However, we forgot to provide the reviewer log-in for PRIDE with which you can check the proteomic data:

Username: reviewer52552@ebi.ac.uk

Password: ors1fxFI

3. Please write out the result for Figure 4e in the Results section.

We wrote out the result for figure 4 e in track changes. We noticed that, after pooling the dataset from fig 2 a with fig 4 e upon previous request of reviewer 2, a significant difference in size of 10 nm was observed between rEV and rEV-PEG. We now corrected this in the manuscript in track changes.

4. Please add (3h) at an appropriate place in text. The order of ELISA and gag-EGFP are not the same in the Results text and in Fig 3.

The order of ELISA and western blot is now changed so that it follows the order of figure 3.

5. "Freezing (for 6 months and 12 months) and thawing of rEV (supplementary figure) followed by storage of rEV at 4C..."

Surely the freeze-thaw experiments and +4C storage experiments were not connected? Please rephrase.

Figure still contains FT1 which in another figure means Flow Through, but here it is a remnant of Freeze-Thaw1 in the previous version?

We rephrased the text in track changes and removed “FT1” from the figure legends. The latter was indeed a remnant of the previous version.

6. This previous query was not addressed: For the following assays data concerning the sample numbers/sample size/used material (volume/particle number/ protein) for each assay would be important information:

Protein mass spectrometry

Lipid mass spectrometry

Total cholesterol

ELISA

Number of repeats (technical) or biological replicate is still missing at least for lipidomics.

We now included the particle number and biological replicates for lipidomics.

Content & writing (major):

1. Introduction:

a. behave similarly

b. Current reference materials...liposomes, which lack EV-like properties?

c. ...are commercially available, but they lack...?

d. Recent approaches have tried...?

e. rEV find their origin in the major structural component of HIV-1 virus particles...?

All suggestions were followed and were made in track changes.

2. Results: The last sentence of the first para ("We evaluated at least three biological replicates of rEV... ") seems to be out of place here, since by 1f nothing has been done to rEV yet as comparisons, which start from Fig 2. Reposition?

We repositioned the last sentence so that it is the beginning of the next paragraph. The changes were made in track changes.

3.

a. Supplementary Fig 2 data addressed before the real Fig 2? Change order?

We did not change the order of sup fig 2 and fig 2 because we think that this is a good beginning of the comparison. In this way we make sure the reader knows that rEV and all EV used for comparison were isolated based on density.

b. p.10 exclusion chromatography is missing from line 13.

We did not include “exclusion chromatography”, but rather changed “ODG centrifugation” to “density” as this fits better in the sentence. All changes were made in track changes.

c. p.11 Suggestion (as plasma behaves differently and is described separately after the other biofluids): replace various with these.

We replaced “various” with “these” in track changes.

d. The sentence “This shift was reversible and inducible” is still in text, although the authors claim to have corrected it. Suggestion: “This shift was reversible, as proteinase K (PK)... “

At the end of the paragraph: “This shift was inducible, as an addition of physiological amounts of IgM...”

We now followed the reviewers suggestion made adaptations in track changes.

4. Why is there on p.12 a reference (19) for ODG centrifugation, but no reference for SEC or differential centrifugation?

We now included an extra reference for SEC and dUC.

5. Discussion:

a. An urge to obtain a biological reference material or A need for a biological ...?

b. To meet this need,

c. a gag-protein -induced

d. rEV were generated...

e. rEV are safe, since

f. rEV are intrinsically stable... What does intrinsically stable mean?

g. In depth analysis for molecular function and biological processes revealed...

All above mentioned corrections were adapted in the manuscript file in track changes.

h. What does this mean: “As such, rEV-PEG do not interfere with downstream analysis of sample EV composition (proteomics, lipidomics, transcriptomics, metabolomics) and function. Please, rephrase.

We rephrased the sentence in track changes so that it is easier to understand.

i. (size and large EV-specific protein versus other)? Other what?

We included some extra information in track changes so that our claim can be better understood.

6. Figures:

a. In Fig 1 the schematic of rEV is very useful. However, as it highlights only some of the sample EV-related molecules shared with the rEV (although many more were discovered by proteomics and lipidomics), it may be helpful for the reader if the exemplary nature of the depicted molecules is somehow stated e.g. "Schematic presentation of rEV showing representative/some of the key/molecular components shared with sample EVs..."

We followed the reviewers suggestion and made this adaptation the figure legend.

b. Supplementary Fig 1d: n=6 refers to technical or biological replicates?

N=6 refers to biological replicates. This is now stated as well in the figure legend.

7. References:

a. Figure 1. The reviewer fully understands the need to explain the biogenesis of rEV in panel b and is convinced that the readers will also appreciate it. However, since none of the information in panel b was generated in this study, it would be appropriate to reference the source for the information.

We followed the reviewers suggestion and now included a reference to a relevant review paper in the figure legend.

Reviewer 3

Since the authors made all the responses to my comments, the manuscript seems good to be published.

We thank the reviewer for the valuable suggestions that were given, which clearly made our claims stronger throughout the manuscript.